# Three dimensional architected thermoelectric devices with high toughness and power conversion efficiency

Vaithinathan Karthikeyan[1,2,7], James Utama Surjadi[3,4,7], Xiaocui Li[3], Rong Fan[3], Vaskuri C. S. Theja [1,2], Wen Jung Li [3], Yang Lu[3,4,5] & Vellaisamy A. L. Roy [6]

For decades, the widespread application of thermoelectric generators has been plagued by two major limitations: heat stagnation in its legs, which limits power conversion efficiency, and inherent brittleness of its constituents, which accelerates thermoelectric generator failure. While notable progress has been made to overcome these quintessential flaws, the state-of-the-art suffers from an apparent mismatch between thermoelectric performance and mechanical toughness. Here, we demonstrate an approach to potentially enhance the power conversion efficiency while suppressing the brittle failure in thermoelectric materials. By harnessing the enhanced thermal impedance induced by the cellular architecture of microlattices with the exceptional strength and ductility (>50% compressive strain) derived from partial carbonization, we fabricate three-dimensional (3D) architected thermoelectric generators that exhibit a specific energy absorption of ~30 J g$^{-1}$ and power conversion efficiency of ~10%. We hope our work will improve future thermoelectric generator fabrication design through additive manufacturing with excellent thermoelectric properties and mechanical robustness.

The increasing global energy consumption scenario and their negative impact on climatic change oblige the development of sustainable energy sources. For most of the system, over two-third of their total energy consumption is released as heat output, about 50 TWh/year is lost as waste heat from industries[1,2]. Harvesting only 1% of the waste heat can account for a reduction of about 0.25 Tg/year in $CO_2$ emissions[1–3]. Solid-state Thermoelectric Generators (TEGs) are one of the best potential alternate sources for direct conversion of waste heat to electricity. Power conversion efficiency ($\eta_{max}$) of TEGs are controlled by the temperature difference ($\delta T$) and thermoelectric material's figure of merit ($zT$)[4].

$$\eta_{max} = \frac{\delta T}{T_{Hot}} \frac{\sqrt{1+z\overline{T}} - 1}{\sqrt{1+z\overline{T}} + T_{Cold}/T_{Hot}} \quad (1)$$

Conventional thermoelectric generators consist of monolithic arrays of p and n-type material legs[1]. Over the past decades, studies focusing on enhancing the performance of thermoelectric materials with defect and dopant perspective[5,6] has led to significant breakthroughs in achieving higher $zT$. Despite this improved material's

[1]Department of Materials Science and Engineering, City University of Hong Kong, Kowloon, Hong Kong. [2]State Key Laboratory for Terahertz and Millimeter Waves, City University of Hong Kong, Kowloon, Hong Kong. [3]Department of Mechanical Engineering, City University of Hong Kong, Kowloon, Hong Kong. [4]Hong Kong Institute for Advanced Study, City University of Hong Kong, Kowloon, Hong Kong. [5]Department of Mechanical Engineering, The University of Hong Kong, Pokfulam, Hong Kong. [6]School of Science and Technology, Hong Kong Metropolitan University, Ho Man Tin, Hong Kong. [7]These authors contributed equally: Vaithinathan Karthikeyan, James Utama Surjadi. ✉e-mail: ylu1@hku.hk; vroy@hkmu.edu.hk

performance, the extensive application of TEGs has been hampered by two main limitations: (1) heat stagnation in p and n-type material legs and (2) inherent brittleness of its constituents. The compact solid nature of its monolithic legs causes the stagnation of heat within, making it a challenge to maintain a constant thermal gradient for the applied heat, which in turn reduces the power conversion efficiency and module lifetime irrespective of the thermoelectric material's performance[6]. Recently, there is a growing demand to feature additive manufacturing (i.e., 3D printing) techniques to prevail over the geometrical limitations of thermoelectric modules[7]. However, most of them are also the same replica of conventional device design with monolithic legs, which is unfavorable for retaining large thermal gradient[7,8]. Utilizing cellular topology with a larger surface-to-volume ratio can potentially provide a sustainable thermal gradient by repairing the heat stagnation in the structure. For instance, Kim et al.[9] recently used direct-ink writing to fabricate three-dimensional (3D) thermoelectric cellular micro-architectures with a woodpile-like structure which exhibited large temperature gradients and large power density. Nevertheless, thermoelectric devices with solid-beam structures composed entirely of $(Bi,Sb)_2(Te,Se)_3$-based thermoelectric materials are still prone to suffer from brittle fracture and low mechanical strength, which can cause difficulties in thermoelectric module assembly and miniaturization. This problem also raises concerns over the TEG module's reliability and longevity[5,6]. While significant efforts were dedicated to undertaking these limitations, state-of-the-art approaches suffer from a conflict that exists between the thermoelectric properties and mechanical toughness. For instance, TEG systems widely reported for their exceptional thermoelectric performance and material stability are extremely brittle (e.g., $Bi_2Te_3$, SnSe)[10–14]. Conversely, other potential TE materials with high ductility are typically inferior in terms of its figure of merit[15–18].

3D microlattice structures with precisely controlled architectures have emerged as a potential solution to access uncharted regions in material property space and provide enhanced load-bearing capabilities (e.g., high strength at low densities) over conventional foams[19–23]. Furthermore, topological optimization methods can accurately tailor not only the mechanical properties of these cellular structures for a specific application but also enhance their ability to sustain large thermal gradients, which are unrealistic with conventional bulk structures. Here, we report a facile approach to overcome the major restrictions in TEGs by developing a core−shell thermoelectric 3D microlattice TEGs consisting of hybrid carbon cores and thermoelectric shells (p-$Sb_2Te_3$ and n-$Bi_2Te_3$) as shown in Fig. 1. By harnessing the synergetic effect between the strong, ductile architected core and low-dimensional shell, we achieved unsurpassed specific toughness and extraordinary power conversion efficiency, outperforming the previously reported monolithic TEGs. This work provides a strategy to overcome the inherent coupling between thermoelectric power conversion and mechanical strength for the creation of next-generation TEGs.

## Results and discussion
### Hierarchical structure of thermoelectric microlattices
3D core−shell thermoelectric face-centered cubic (FCC)-like microlattices (composed of diagonal struts from the edges of the cubic unit cell intersecting at the center of the cubic faces) are produced by fabricating polyethylene glycol diacrylate (PEGDA) polymer microlattices via DLP 3D printing (Fig. 1a). To enhance the high-temperature thermal stability of the polymer microlattice cores, the polymer cores were thermally annealed in the proximity of its decomposition temperature (Supplementary Fig. S1b) under controlled time and environment to achieve a uniform degree of partial carbonization throughout the lattices. The composition of the resulting partially carbonized PEGDA (PC-PEGDA) or hybrid carbon composite microlattices was analyzed via Fourier transform infrared (FTIR)

spectroscopy, X-ray photoelectron spectroscopy (XPS), and Raman spectroscopy (Supplementary Fig. S1c–e), where the intensity of polymer peaks reduces with the evidential emerging of D-band and G-band carbon peaks[24–27]. The rise in C:O ratio observed in the XPS spectra implies the formation of pyrolytic carbon in the polymer microlattice to form a hybrid carbon composite[28–30]. Consequently, p-type $Sb_2Te_3$ and n-type $Bi_2Te_3$ thin films were deposited onto the partially carbonized microlattices by thermal evaporation to develop core−shell thermoelectric microlattices, as illustrated in Fig. 1b. These thermoelectric microlattices are electrically connected in series and assembled into a novel 3D TEG device (Fig. 1c). Figure 1d is an illustration of a representative unit cell of the thermoelectric core−shell microlattice. Figure 1e–i shows the hierarchical structure of a core−shell TE microlattice, exhibiting feature sizes ranging from millimeters to nanometers. 3D core−shell thermoelectric microlattice consists of a periodic arrangement of $4 \times 4 \times 4$ FCC-like lattices with unit cell size of ~1.25 mm (Fig. 1e, f). Figure 1g exhibits the circular cross-section of the core−shell microlattice strut, showing the ductile hybrid carbon core and the thermoelectric shell. Scanning electron microscopy (SEM) image of the cross-section reveals a uniform deposition, with the film thickness measured to be ~1 μm (Fig. 1h). The transmission electron microscopy (TEM) image of the thermoelectric shell is shown in Fig. 1i, showing an average grain size of ~20 nm, indicative of a nanocrystalline structure. The elemental composition and stoichiometry of the thermoelectric layer on the core−shell microlattices were analyzed via energy-dispersive X-ray (EDX) mapping (Supplementary Fig. S2). The microstructures of the deposited TE shell were also investigated via X-ray diffraction (XRD), XPS, and high-resolution TEM (HRTEM), as shown in Supplementary Figs. S3–5.

### Thermoelectric properties of 3D core−shell microlattices
Temperature-dependent thermoelectric properties of 3D microlattice architected p-n type legs are demonstrated here with a high degree of processable and reproducibility. TE properties are measured for 1 μm thickness of p- and n-type compact and intact films coated over the 3D scaffold microlattices. Seebeck coefficient of the 3D p-$Sb_2Te_3$ and n-$Bi_2Te_3$ ranges between 160–240 μVK⁻¹ and 120–130 μVK⁻¹ in their operating temperature range, respectively. The electrical conductivities of the 3D structures increase with the temperature following the Petritz-mobility model[31,32] for 3D $Sb_2Te_3$ and $Bi_2Te_3$, the electrical conductivity ranges between 100–250 S cm⁻¹ and 360–435 S cm⁻¹, respectively (Fig. 2a, b). However, the power factor remains consistent with peak values of 6.4 and 7 μW cm⁻¹ K⁻² for 3D $Sb_2Te_3$ and $Bi_2Te_3$, respectively, as represented in Fig. 2c. For a better understanding of the electrical properties, the temperature-dependent charge carrier concentration and mobility were determined using Hall effect measurement system for both p- and n-type samples, where the synergic increase in carrier transport between the nanograins explains the increase in electrical conductivities with temperature (Supplementary Fig. S6)[33,34]. The charge carrier concentration for p- and n-type samples at room temperature were $1.09 \times 10^{20}$ cm⁻³ and $9.89 \times 10^{19}$ cm⁻³, respectively. The electrical properties of the deposited thin films are comparable with their bulk counterparts, which helps to exhibit an equivalent power factor as the bulk samples[34]. Temperature-dependent thermal conductivity of p- and n-type thin film samples ranges between 0.23−0.45 W m⁻¹ K⁻¹ in the operating temperature region (Fig. 2d), which is almost an order of magnitude lower than their bulk counterparts. As per the Callaway's model, under relaxation time approximation, the phonon relaxation time purely depends on the grain size and thickness of the thin film[35,36]. Thus, here the thermal conductivity of thin films is controlled by nano-textured grains rather than the strain-induced effect in bulk materials. The temperature dependence of the thermal conductivity must be attributed by the increasing carrier transport. The partially carbonized (amorphous carbon) scaffold inherits a lower thermal conductivity,

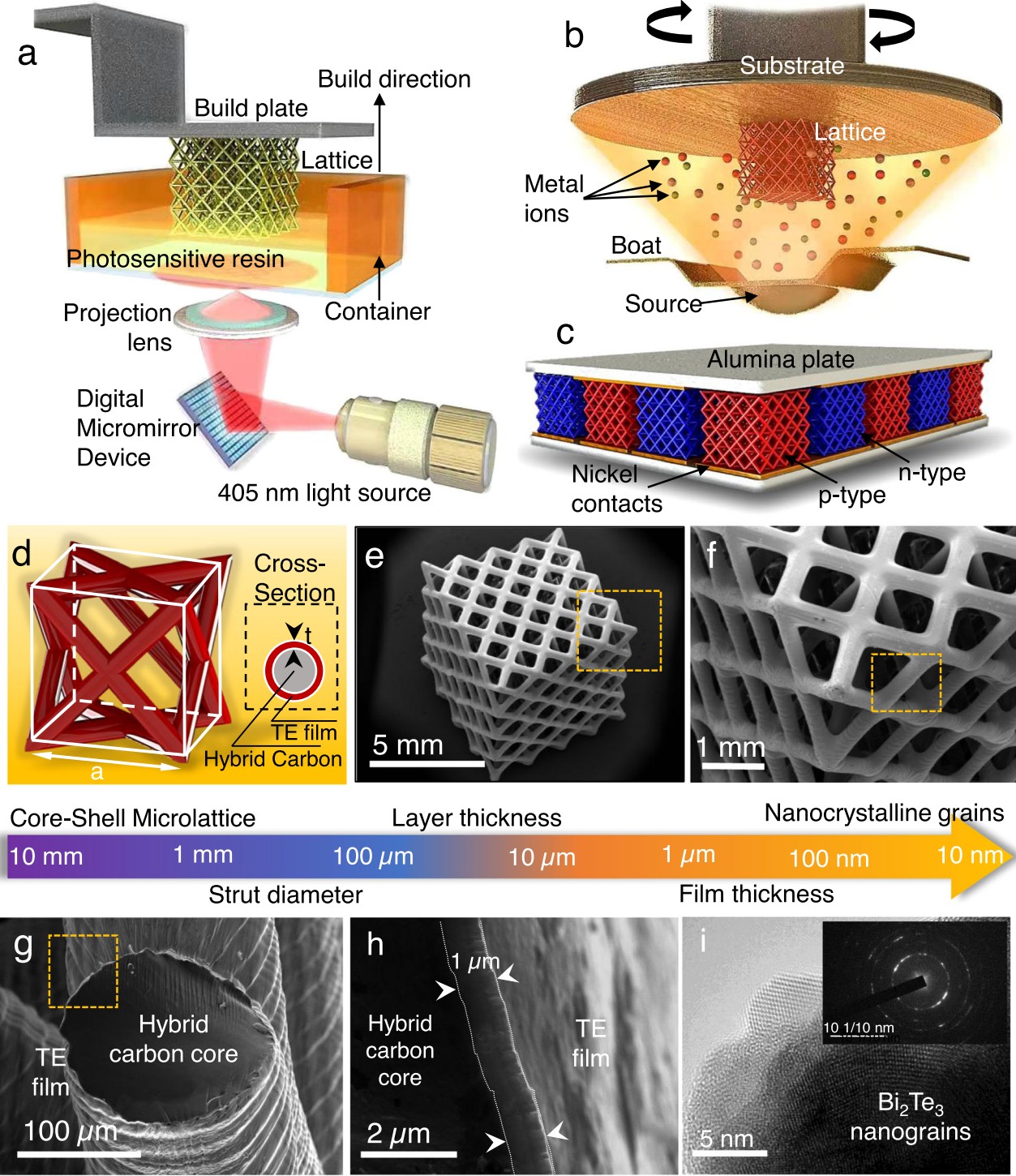

**Fig. 1 | Fabrication of 3D architected thermoelectric generators (TEGs).**
**a** Schematic illustration of the Digital Light Projection (DLP) 3D printing technology used to fabricate the polymer microlattices, which were then partially carbonized to produce tougher partially carbonized (amorphous carbon) microlattices. **b** Illustration of the thermal evaporation process used to deposit the partially carbonized microlattices with thermoelectric (TE) films of antimony telluride (p-$Sb_2Te_3$) and bismuth telluride (n-$Bi_2Te_3$). **c** Illustration of an assembled 3D architected TEG consisting of alternating p-and n-type microlattice legs connected in series. **d** Representative unit cell of the thermoelectric microlattice consisting of a core–shell structure with partially carbonized PEGDA as its core and thermoelectric film as the shell to produce the tough, ductile TEGs. **e, f** Hierarchical structure of the thermoelectric microlattice fabricated, showing critical feature sizes ranging from a few millimeters to several nanometers. **g** Cross-section of the amorphous carbon beam of the 3D microlattice architected TE legs. **h** Demonstrates the ductile core and the deposited TE film. **i** Nanograin structure of the deposited TE films.

which implies that it does not contribute to the total thermal conductivity of the samples. Overall, the lower thermal conductivity and high power factor result in a larger figure of merit (zT) values of 0.97 and 1.09 for p- and n-type at 550 K, respectively (Fig. 2e). These high zT values are made possible due to the higher temperature difference maintained by the novel 3D structured TEG leg designs. The obtained thermoelectric performance of the p- and n-type legs were checked for repeatability with multiple test samples.

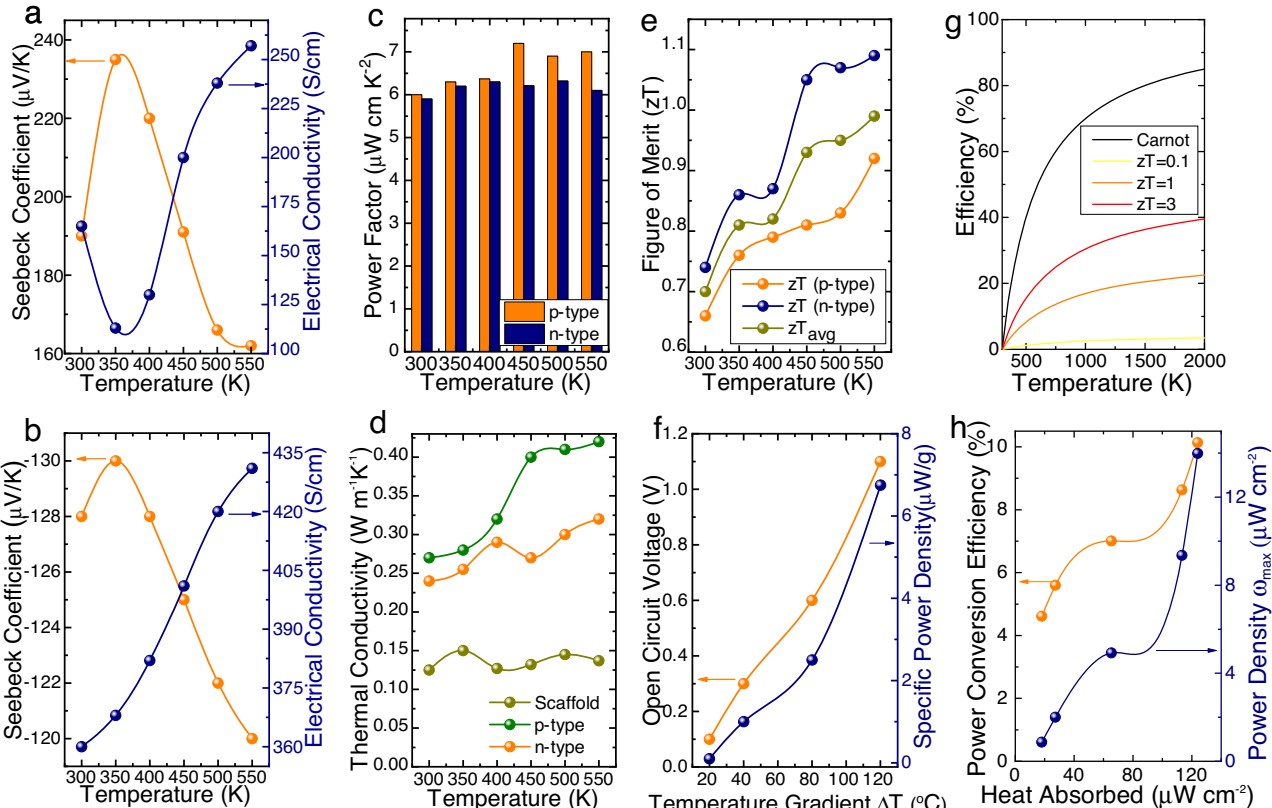

**Fig. 2 | Thermoelectric properties of the architected TEGs. a**, **b** Seebeck coefficient and electrical conductivity of the 3D p-$Sb_2Te_3$ and n-$Bi_2Te_3$ legs with respect to operating temperature. **c** Power factor of the 3D architected p-$Sb_2Te_3$ and n-$Bi_2Te_3$ thermoelectric legs with temperature. **d** Temperature-dependent thermal conductivity of p- and n-type thin films and the amorphous carbon core with respect to temperature. **e** Figure of merit (zT) p-$Sb_2Te_3$, n-$Bi_2Te_3$ legs at different temperatures, evidencing the effect of partially carbonized core and architecture in enhancing the zT compared to previously reported works. **f** Maximum open-circuit voltage and specific power density of the 3D architected TEG. **g** Figure of merit comparison with Carnot engine efficiency. **h** Device efficiency and power density of the TEG within their operating temperature range.

## High strength and ductility of 3D microlattice TEGs

During the real-time operating environment, the applied external load on the thermoelectric module is mostly compressive in nature. Intensifying the compressive strength of the materials through a 3D novel cellular topological model aids to higher operational efficiency in the TE module. Most reported and commonly used commercial bulk $Bi_2Te_3$-based thermoelectric modules suffer brittle fracture at low compressive strains within 5%[8]. Here, we demonstrate loading-unloading compression analysis on the topological microlattice structures which exhibits nearly 100% elastic recovery up to strain levels of 10% (Fig. 3a). Our results imply negligible deformation and ensure thermoelectric operation of microlattice structures even at much higher strains than the allowable strain of conventional monolithic modules.

To analyze the mechanical properties of our thermoelectric microlattices under extreme loading conditions, we further conducted in situ uniaxial compression tests up to 50% strain on the thermoelectric microlattices. Figure 3b and d exhibits the stress–strain curve and deformation behavior of the structures at different stages, respectively. As shown in Fig. 3b, the thermoelectric microlattices with the hybrid carbon acting as its core, and thermoelectric thin film as its shell exhibits significantly enhanced compressive modulus of ~450 MPa and strength of ~35 MPa compared to the polymer microlattices. Furthermore, the thermoelectric microlattices exhibit localized plastic buckling without any apparent strut fracture upon yielding as shown in Fig. 3d. The suppression of strut fracture at large compressive strains (~50%) can also be verified through the stress–strain curve, indicated by the absence of any large, drastic

stress drops or serrations. Subsequently, a layer-by-layer deformation of the microlattice via buckling followed by densification demonstrates the characteristic of highly elastic/plastic lattices[37]. This proves the excellent ductility and energy absorption capabilities of the thermoelectric microlattices. Note that the thermoelectric microlattices retained most of their electrical properties even after being subjected to large deformations up to 75% strain, further demonstrating the mechanical robustness of the microlattice (Supplementary Fig. S7). Similar mechanical characteristics were observed for the uncoated partially carbonized microlattices as well with a slight decrease in specific strength and modulus as shown in Supplementary Fig. S8. On the other hand, the pristine polymer microlattice experiences localized Euler buckling of its struts, which resulted in rotation and deformation at its nodes. This influence is demonstrated by the observed stress plateauing in stress–strain curve. At higher compressive strains, strut fracture occurs primarily at the nodes due to stress concentration, which is the typical location of fracture in lattices[38,39].

As a result of the controlled pyrolysis process, a substantial increase in compressive strength is achieved by the core, resulting in the thermoelectric composite microlattices to exceed the specific strength and ductility of its commercially available bulk thermoelectric counterpart. This strength of the microlattice struts is derived from the formation of pyrolytic carbon consisting of interconnected curved graphene fragments[29] and the presence of remaining polymer chains in the hybrid composite core act as an energy dissipation matrix that restricts the brittle shear fracture of the carbon fragments, giving rise to the high ductility[40]. The quantitative comparison of differences between pure polymer, partially carbonized polymer, and

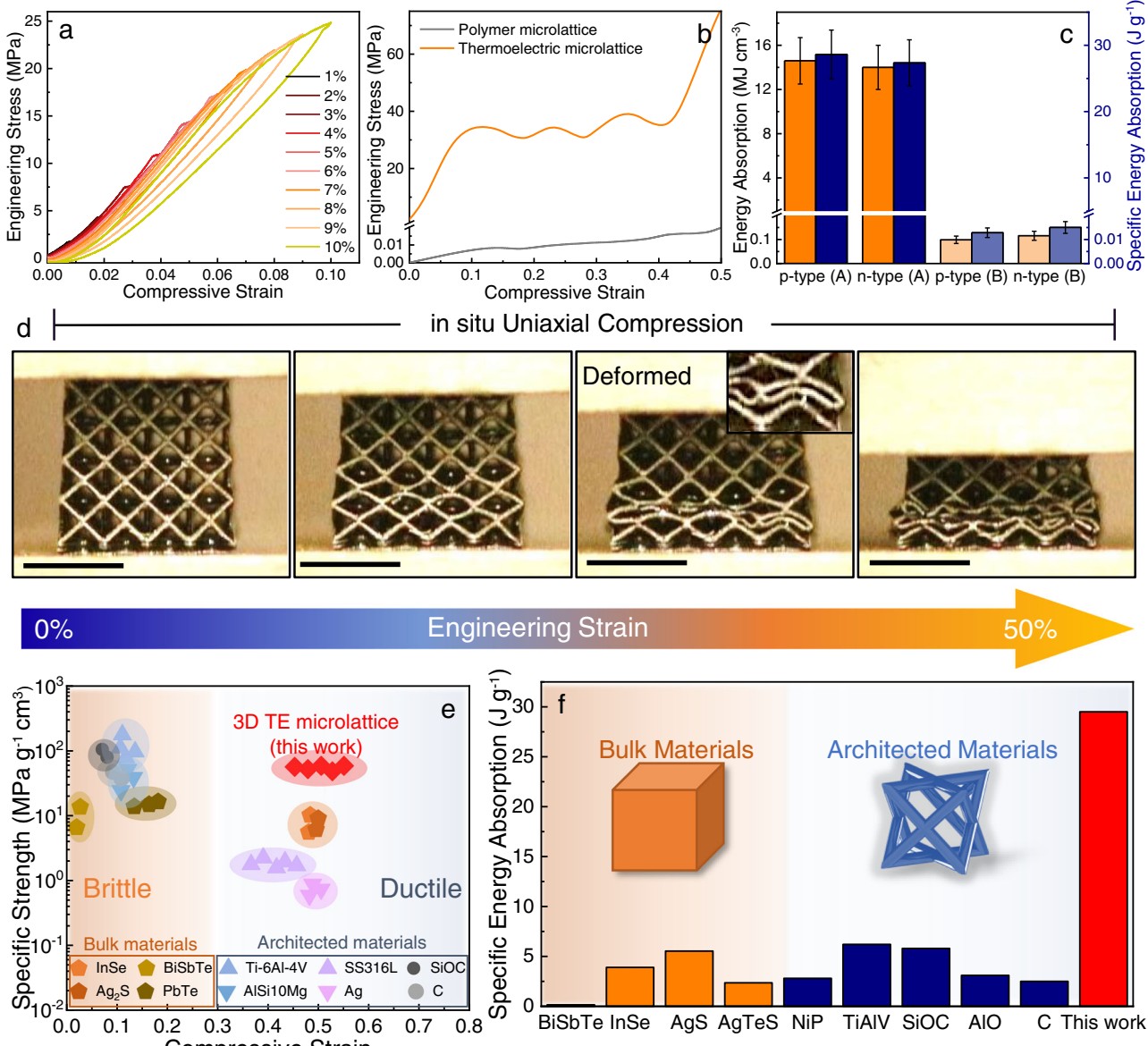

**Fig. 3 | In situ mechanical characterization of thermoelectric microlattices.**
**a** Loading-unloading compression curves of the thermoelectric microlattice showing the near-fully elastic behavior at strains higher than most of previously reported brittle TEGs and microlattices. **b** Stress−strain curves from uniaxial compression of the thermoelectric microlattices and polymer microlattices, showing the drastic increase in mechanical strength (>2 orders of magnitude) of the polymer microlattice upon partial carbonization and coating with thermoelectric film. **c** Energy absorption capabilities of the developed thermoelectric microlattices compared to the commercially obtained monolithic TEG legs under uniaxial compression. Data are presented as mean while the error bar represents the standard deviation. **d** Deformation behavior of the thermoelectric microlattices under uniaxial compression, demonstrating its exceptional ductility with negligible strut

fracture at high compressive strains (scale: 5 mm). **e** Specific strength versus compressive strain of our thermoelectric microlattice compared to previously reported monolithic TEGs and microlattices. In this case, compressive strain was taken as either the strain at which catastrophic/brittle fracture occurs (i.e., large stress drop) for the samples in the brittle region or the maximum reported strain applied (for ductile samples). The referenced data were extracted from the following: InSe[18], Ag₂S[15], BiSbTe[8], PbTe[41], Ti-6Al-4V[42], AlSi10Mg[43], SS316L[44], Ag[45], SiOC[46], and C[47]. **f** Specific energy absorption (SEA) of thermoelectric microlattices compared to previously reported monolithic semiconductors, TEGs, and microlattices. The referenced data were extracted from the following: BiSbTe[8], InSe[18], AgS[15], AgTeS[16], NiP[49], TiAlV[42], SiOC[46], AlO[48], and C[47].

thermoelectric microlattices can be found in Supplementary Fig. S8. Thus, the controlled carbonization of the polymer microlattices to produce hybrid carbon composite core−shell thermoelectric microlattice not only demonstrates enhanced thermal stability to withstand the operating temperatures of TEGs but also increases both its strength and toughness (over 100 times). A more detailed discussion on the dominant deformation mechanism of the thermoelectric microlattices is shown in Supplementary Information and Supplementary Fig. S9. We compared the mechanical properties of our

core−shell thermoelectric microlattices against previously reported monolithic TEGs[8,15,18,41] and architected materials[42–49], as shown in Fig. 3e, f. Our TE microlattice structure outperforms bulk TE materials and other ductile lattice structures (e.g., stainless steels[44], Ag[45]) in terms of its strength per unit density by several times while maintaining exceptional ductility. Previous reports on micro/nanolattices with comparable specific strengths characteristically exhibit either catastrophic or brittle failure (e.g., silicon carbides[46], Ti-alloys[42], Al-alloys[43]) at relatively low compressive strains (~20%). Owing to the high

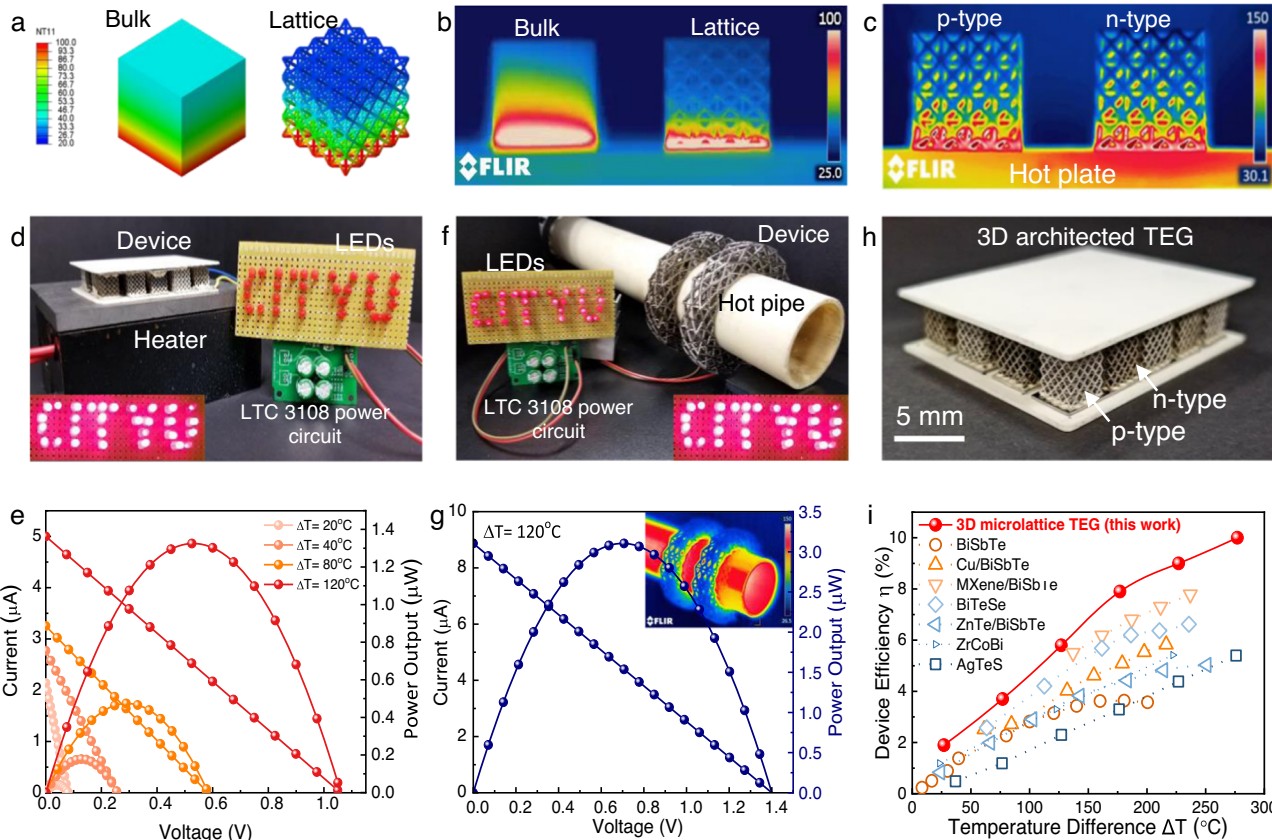

**Fig. 4 | Real-time applications of 3D architected TEGs. a** Heat transfer model of the conventional solid TEG legs and microlattice TEG legs, showing the reduced effective thermal conductivity of the microlattice legs compared to monolithic legs. **b** IR image of the solid and architected TEG legs verifying the simulated model. **c** IR image of the p-type and n-type architected TEG legs under operating conditions. **d**, **e** Demonstrating the ability of the developed 3D architected TEGs (planar and circular) to conform to any desired surface and power electronic circuits (i.e.,

light up LEDs). **f** Zoomed-in optical image of the planar architected TEG developed. **g**, **h** Load test results of the 3D architected TEGs (planar and circular, respectively). **i** Comparison of the power conversion efficiency (PCE) of our architected TEG compared to previously reported TEGs at various temperatures. The referenced data were extracted from the following: BiSbTe[51], Cu/BiSbTe[51], MXene/BiSbTe[55], BiTeSe[14], ZnTe/BiSeTe[54], ZrCoBi[56], AgTeS[16].

specific strength and ductile deformation behavior, our thermoelectric core−shell microlattices possess over 100 times higher specific energy absorption compared to $Bi_2Te_3$-based TEG legs[8] and higher than most previously reported bulk TEG legs[15,18,41]/microlattices[42–48,50], indicating that our architected TEGs possess exceptional combination of strength and ductility at low density.

**Thermoelectric device performance and potential applications**
To demonstrate a thermoelectric generator with the proposed microlattice design structure, ten pairs of p- and n-type microlattice structured legs are connected in series with nickel contacts and enclosed between the insulating ceramic plates as illustrated in Fig. 1c and shown in Fig. 4 h. The power conversion capacity of the constructed microlattice TEG device was analyzed and discussed in detail by calculating the heat absorbed by the 3D core−shell TE microlattice device. The electrical performance of the microlattice TEG device is observed at the temperature difference of 20 °C, 40 °C, 80 °C, and 120 °C between the hot and cold sides, respectively. Under the no-load condition, the experimentally measured maximum open-circuit voltage of the TEG was 1.1 V at ΔT of 120 °C (Fig. 2f). The maximum output power density of the microlattice TE device was calculated as ~14 μW cm⁻² (Fig. 2h). Specific power density (i.e., power to mass ratio) for the TEG was calculated as ~7 μW g⁻¹, which is in good comparison to the conventional bulk $Bi_2Te_3$ TEG device[51–53]. These excellent electrical properties and the ability of the TEG to maintain a larger temperature difference adds up to a total device efficiency of ~10% (Fig. 2h). The

power density comparison of our results with previous reports of 3D structured TE devices are tabulated in Supplementary Table S1, which illustrates our 3D microlattice results are in close comparison to commercial TE device. The heat transfer comparison finite element model (FEM) between the conventional solid TEG legs and the proposed microlattice TEG legs are shown in Fig. 4a and verified via IR imaging in Fig. 4b. Figure 4a, b demonstrates the capability of the microlattice structure to naturally maintain a significantly larger temperature difference between hot and cold sides compared to its bulk counterpart. The slight difference in thermal conductivity between the p- and n-type TEG microlattice legs were also verified through IR imaging (Fig. 4c), where the n-type exhibits a slightly higher thermal gradient due to its lower thermal conductivity. In Fig. 4b and c, we attribute the hotter areas in the infrared images to the difference in thermal conductivity of the samples. For instance, in Fig. 4b, the bulk material conducts heat slower than the microlattice sample, giving rise to hotter area near the bottom surface of the bulk sample. Similarly, the thermal conductivity of the n-type thermoelectric material is lower than that of the p-type material, which results in the area near the bottom surface of the n-type thermoelectric microlattice being hotter than the area at the bottom of the p-type sample (Fig. 4c). The real-time performance of the microlattice TEG under load condition with varied temperature differences was investigated in detail. The microlattice TEG was placed over a resistive heat source, and the temperature difference was monitored using a K-type thermocouple. The cold side of the TEG was naturally cooled by atmospheric airflow instead of any

forced heat transfer mechanisms such as water-cooled or heat sinks (Fig. 4d). The load test at specified ΔT was conducted using variable resistive loads; the output characteristics of microlattice based TEG is shown in Fig. 4e, where the maximum power output of ~1.4 μW. The obtained maximum power output of the TEG is in good agreement with its calculated specific power density. To demonstrate a conformable 3D structural design to the specific shape of heat source, we have also constructed and studied the performance of a circular TEG design conformable to tubular heat sources like chimneys and vehicle exhausts. The circular-shaped 3D printed TE structure was mounted on an alumina pipe with a hot air exhaust outlet of 150°C, as shown in Fig. 4f. Temperature difference of 120°C was maintained between the circular 3D TE structures to conduct the load test, which exhibited a maximum power output of 3.2 μW (Fig. 4g). The maximum power point in each TEG was tracked and stored in a LTC3108 power circuit, which can light up the LEDs, as shown in the inset of Fig. 4d, f. Throughout the experiment, the cold side of the TEG was found to remain at room temperature, demonstrating the ability of our microlattice TEGs to practically maintain significant thermal gradient and power output. We further compared the power conversion efficiency of our architected TEG in Fig. 4d with previously reported TEGs (Fig. 4i)[8,14,16,51,54–56]. Owing to the drastically reduced thermal conductivity caused by the synergy between the open cellular architecture and quantum confinement effect due to the reduced dimensions, our device efficiency surpasses all other TEGs. With this level of device efficiency, thermoelectric power conversion technology can stand comparable with any other renewable energy technologies. To elucidate the performance effectiveness of the 3D core−shell TE device, we performed a simulation study based on 3D TE finite element model (FEM) as shown in Supplementary Fig. S10. Supporting the real-time experimental results, the simulated thermal distribution, and the voltage generation from the 3D TE core−shell microlattice device demonstrates a larger thermal gradient at higher operating temperature range. Moreover, the simulated power output characteristics validates our obtained experimental results demonstrating a maximum power output of 1.5 μW at ΔT of 120 °C.

In this work, we demonstrate a strategy to sustain large thermal gradient in TEGs and potentially suppress the inherent brittleness. With a core−shell microlattice configuration combining partially carbonized composite 3D architectures with thermoelectric thin films, we have fabricated TEGs with microlattice geometries that could conform over any heat-radiating surface, with extreme toughness and power conversion efficiency. This proposed 3D thermoelectric microlattice architecture facilitates larger electrical conductivity and low thermal conductivity in the device, which results in a higher power conversion efficiency in comparison to commercial devices. Furthermore, the enhanced toughness is attributed to the carbon composite core which possess multi-fold higher specific strength than previously reported bulk TE materials and engineering alloys such as stainless steels, as well as drastically improved fracture resistance that can resist more than ten times deformation than its pure monolithic TE counterpart (Bismuth Telluride - based alloys). The boundless design space of 3D architected TEGs will open to a new standard of thermoelectric devices that facilitate macro- to microscale process ability for multi-scale waste heat recovery and wearable applications.

## Methods

### Fabrication and characterization of architected thermoelectric devices

Initially, polymer lattices were fabricated using a DLP 3D printer (Micromake L2) with 405 nm light source. The photosensitive resin consists of 98 wt% polyethylene glycol diacrylate (PEGDA) Mw 700 and 2 wt% phenyl bis(2,4,6-trimethyl benzoyl)phosphine oxide (BAPO) photoinitiator. The slicing distance was set at 50 μm, with a curing time of 8000 ms for each 2D layer. The 3D geometry of the polymer lattices was chosen to be $4 \times 4 \times 4$ unit cells. This serves to allow the lattice to behave as a material by minimizing edge effects. Considering the experimental practicalities, FCC structure was explicitly designed for this work to optimize the openness of the unit cells for thermoelectric film deposition while maintaining a reasonably rigid and weight-efficient geometry for structural applications owing to its partially stretching-dominated nature. Subsequently, the PEGDA polymer lattices were partially carbonized at 350 °C for 4 h in inert environment. The thermoelectric films, namely $Sb_2Te_3$ and $Bi_2Te_3$ with a thickness of about 1 μm were deposited on the partially carbonized PEGDA (PC-PEGDA) or hybrid carbon microlattices at room temperature by thermal evaporation of high purity (99.99%) as p-type and n-type legs, respectively. The deposition was performed at a working pressure of ~$10^{-7}$ mBar and a deposition rate of ~10 Å/s and ~15 Å/s for $Sb_2Te_3$ and $Bi_2Te_3$, respectively. From the thermally deposited p- and n-type 3D TE microlattices, a thermoelectric device with 10 pairs of p-n TE legs. The p-n legs are connected in series with nickel electrodes with the use of Ag-Sn solder paste with thin insulating ceramic plate as thermal contacts as shown in Fig. 1. Further, the fabrication 3D TE device assembly was annealed at 220°C to establish mechanical stability to the assembly by soldering the structure between the ceramic plates.

XRD pattern of the deposited $Bi_2Te_3$ and $Sb_2Te_3$ thin films indicates the phase purity and crystallinity (Supplementary Fig. S3a, 3b). The XRD patterns suggest the formation of a rhombohedral crystal of $Bi_2Te_3$ with a dominant characteristic peak (015) (R3m, JCPDS card #15-0863) and rhombohedral crystal structure of $Sb_2Te_3$ with the orientation along (009), (015) (R3m, JCPDS card #49-1713). The grain size of the deposited thin films was evaluated by Debye−Scherrer's equation (D = kλ/B cosΘ), and the values are 18.19 and 20.33 nm for $Bi_2Te_3$ and $Sb_2Te_3$, respectively. The lattice constants calculated from the XRD pattern are 10.697 and 10.745 Å for $Bi_2Te_3$ and $Sb_2Te_3$, respectively, which are in good agreement with the standard values. The smaller grain size ensures the formation of uniform and imperfection-free crystalline thin film deposition. For deep structural understanding of the deposited thin metal tellurides, X-ray photoelectron spectroscopy measurements were subsequently performed to further analyze the chemical stoichiometry of the samples using XPS, PHI Model 5802. In the X-ray photoelectron spectra shown in Supplementary Fig. S3c–h, only Bi and Te peaks are resolved with no appearance of extra peaks of carbon and oxygen. This lack of other core level signatures indicates contaminants free clean surface. The surface chemical stoichiometry was analyzed by calculating the area under the Bi $4f_{5/2}$, Bi $4f_{7/2}$, Sb $3d_{5/2}$, Sb $3d_{3/2}$ and Te $3d_{5/2}$, Te $3d_{3/2}$ peaks. The ratio of Te/Bi and Te/Sb is determined as 1.48 ± 0.02 and 1.492 ± 0.02, respectively[57,58].

The surface morphology and elemental composition were analyzed with FESEM, FEI Quanta FEG450 with Energy-Dispersive X-Ray Spectroscopy for elemental analysis and stoichiometry determination. Presence of the nanograins thermoelectric films were analyzed using Transmission Electron Microscopy (TEM, JEOL JEM 2100 F) and selected area electron diffraction were used to analyze the nanostructural properties. Seebeck coefficient of the p- and n-type TE shell-coated 3D microlattice were calculated from the slope of the voltage−temperature difference curve. The electrical conductivity of the TE shell-coated 3D microlattice was calculated at each testing temperature by applying different current values to the sample and the resulting voltage values were measured[59]. For calculating the electrical conductivity, the area of TE shell was calculated from known surface area of the 3D microlattice and wall thickness. Temperature-dependent thermal conductivity were performed by Netzsch LFA 467 Hyperflash. The temperature-dependent measurement of the electrical conductivities of p- and n-type thermoelectric films were performed through Lakeshore Hall measurement method.

## Mechanical testing

The experimental setup for the in situ uniaxial compression tests was conducted at room temperature on the MTS RT/30 electro-mechanical material testing system controlled by TestWorks 4.0 software. A high-speed video camera (Canon EOS-1D X Mark II) equipped with a tele-photo macro lens (Canon EF 100–400 mm f/4.5–5.6 L IS II USM Lens with 77 mm 500D close-up lens attachment) was used to observe the deformation behavior of the lattices. Uniaxial compression tests were performed on the microlattices at a prescribed strain rate of $10^{-3}\,\text{s}^{-1}$. Nanoindentation technique was used to measure the hardness and modulus of thermoelectric thin films by using a TI950 triboIndenter (Bruker) with a standard Berkovich tip under the displacement-controlled mode with a constant strain rate of $0.01\,\text{s}^{-1}$ at room temperature. By using the load-displacement curves from the compression tests, nominal cross-sectional area and total height of the lattice structures, the engineering stress and strain were calculated. The compressive strength of the polymer and thermoelectric microlattices was determined by the first applied peak load observed in the stress–strain curve before failure. The Young's Modulus were measured by fitting the linear elastic region of the stress–strain curves and energy absorption per unit volume was calculated by integration of the area under stress–strain curves. Compressive strain corresponds to the change in length parallel to loading direction, ΔL, divided by the total length/height of sample, L, which is unitless. For specific strength and modulus, the densities of the composite microlattices were calculated by multiplying the relative densities of the lattices with the densities of the solid composite material, calculated by the following equation:

$$\rho_{TE-c} = \frac{\rho_{TE} V_{TE} + \rho_c V_c}{V_{TE} + V_c} \tag{2}$$

where $\rho_{TE}$ and $\rho_{poly}$ represent the estimated densities of the thermoelectric film and hybrid carbon core, respectively. $V_{TE}$ and $V_{poly}$ corresponds to the volumes of the thermoelectric film and hybrid carbon core, respectively. The hardness and modulus of the thin films were measured and calculated using the classic Oliver–Pharr method[60]. To exclude the substrate and indentation size effects on the hardness of the thermoelectric thin films, the hardness was determined at a series of indentation depth-to-film thickness ratios, which is a method previously adopted by other works on thin films. Furthermore, to quantify the substrate effects, the elastic moduli of thermoelectric films were calculated by the following[61]:

$$\frac{1}{E_r} = \frac{2\sqrt{A}}{S\sqrt{\pi}} = \frac{1-v_i^2}{E_i} + \frac{1-v_{TE}^2}{E_{TE}}\left(1 - e^{-\gamma(h-D)/\sqrt{A}}\right) + \frac{1-v_s^2}{E_s}\left(1 - e^{-\gamma(h-D)/\sqrt{A}}\right) \tag{3}$$

where $E_r$ is the reduced elastic modulus, $S$ is the contact stiffness, the slope of initial unloading curve, and $A$ is the projected contact area. $E_i = 1141$ GPa and $v_i = 0.07$ represents the modulus and Poisson's ratio of the Berkovich diamond indenter, while $E_{TE}$ and $v_{TE} = 0.33$ are the modulus and Poisson's ratio of the thermoelectric films, respectively. $E_s$ and $v_s$ are the modulus and Poisson's ratio of the substrate. $h$ is the film thickness, $\gamma$ is a functional parameter, and $D$ represents indentation depth. A minimum of seven indents separated from each other by at least 50 μm was performed on each sample to obtain an average hardness at various indentation depths.

## Thermoelectric device performance analysis

A Keithley 6517A Electrometer was used to measure the thermoelectric power generation and thermal gradient across the hot and cold junction using a K-type thermocouple. A FLIR E33 series infrared thermal imaging camera was used to demonstrate the temperature difference with an accuracy of up to $\pm 2\,^{\circ}$C. The temperature-dependent TEG load characteristics were obtained by Keithley 6517A controlled by Labtrace 2.0 software. To demonstrate the real-time application of the 3D shape conformable TEG, two semicircular TEG microlattice devices were connected in series over a hot air alumina tube with the temperature of 300 °C and then their corresponding power output was recorded using a multimeter. The total power conversion efficiency with respect to heat absorbed in the 3D TE microlattice device is calculated using the following method[62]:

$$\text{Maximum power output,} P_{\max} = \frac{V_{oc}^2}{4R_i} \tag{4}$$

$$\text{Power density,} \omega_{device} = \frac{P_{\max}}{A_{module}} \tag{5}$$

$$\text{Heat conduction in the device,} Q_{out} = \kappa.A.\left(\frac{dT}{dx}\right) \tag{6}$$

Total power conversion efficiency of the device,

$$\eta = \frac{P_{\max}}{Q_h} = \frac{P_{\max}}{P_{\max} + Q_{out}} \tag{7}$$

Where $V_{oc}$ is the open-circuit voltage, $\kappa$ is thermal conductivity of the device, $\frac{dT}{dx}$ is the ratio of temperature difference and device thickness, $Q_h$ is the absorbed heat and $Q_{out}$ is the released heat, respectively. Using this relation, we have performed this calculation using the heat flow and the maximum power output in the microlattice structure, as shown in the Fig. 2h[56,62]. Thermoelectric power conversion performance comparison with previously reported data is tabulated in Supplementary Table 1[7,9,63–70].

## Simulated study of thermal gradient and power output

We performed a finite element study using COMSOL Multiphysics for 3D architected TE generators (Supplementary Fig. S10). Our model calculates the thermal gradient profile of the 3D core–shell TE microlattice structures and full TE device integrated in between a pair of flat alumina plates. During the simulation the cold side and the ambient air was maintained at 25 °C, with natural convective heat transfer coefficient of air as 10 W m² K⁻¹. On the basis of mircolattice geometry and material properties, the internal electrical resistance was estimated as 0.2 kΩ per TE pair at room temperature. The device geometry was maintained as the same experimented TE module dimensions with 10 TE leg pairs with height of 8 mm. Simulation for analyzing the difference in thermal conduction with and without TE shell coating over the polymer core was performed as shown in Supplementary Fig. S10a, which demonstrates a negligible parasitic heat transfer between polymer core and TE shell of the 3D microlattice. Supplementary Fig. S10b, S10c demonstrates the thermal gradient distribution over the device at ΔT of 120 °C. Simulated thermoelectric voltage distribution profile of the 3D device is shown in Supplementary Fig. S10d–f. The power generation at maximum temperature gradient of 120 °C are shown in Supplementary Fig. S10g.

## Data availability

The research data associated with this manuscript are provided in the supporting information and in the source data file. Source data are provided with this paper.

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

## Acknowledgements
The authors gratefully thank the funding supports from the Environmental Conservation Fund (ECF 44/2014) on "waste heat recovery" (V.A.L.R.), Research Grant Council of HKSAR project number T42-717/20 (V.A.L.R. and W.J.L.). Y.L. and J.U.S. acknowledge the funding support from Hong Kong Institute for Advanced Study (HKIAS) and City University of Hong Kong ARG 9667226.

## Author contributions
V.K. and J.U.S. fabricated the samples/devices, performed experiments, analyzed data, and wrote the manuscript; X.L. performed TEM characterization; V.C.S.T. and R.F. performed heat transfer simulation; V.A.L.R., Y.L., and W.J.L. secured the funding support; V.A.L.R. supervised the project on TEG studies while Y.L. on mechanical studies. All authors contributed to the manuscript revision, read, and approved the submitted version.

## Competing interests
The authors declare no competing interests.
