## [Peer Review File · Nature Communications]

REVIEWER COMMENTS

Reviewer #1 (Remarks to the Author):

This publication on 3D architected Thermoelectric Devices shows an innovative concept for the fabrication of future thermoelectric devices. The manuscript will attract broad interest but the thermoelectric characterisation of the 3D device is not mature. The publication claims two high promises for the device with ultrahigh toughness and ultrahigh power efficiency. The first promise is proven by the mechanical testing of the device but the second claim of 10% conversion efficiency is not proven by this manuscript from my point of view. The following aspects are unclear and should be addressed in a revised manuscript:

- 1) Evaporation of metals and semiconductors is ideal for the thin film coatings of planar substrates. How homogenous and conformal is the coating in the center of the 3D architecture. The authors should prepare a cross-sectional cut and investigate this in detail by scanning electron microscopy.
- 2) In figure 2 in many graphs it is not clear, which graph belongs to left and right scale?
- 3) Is the polymeric network structure completely electrical insulating? Does the polymeric network exhibit a Seebeck coefficient without the thermoelectric film? What is the resistance over temperature?
- 4) Fig 2a and 2b: How was the electrical conductivity determined and what is the assumption about film thickness and film area on the network?
- 5) Was the Hall effect determined on planar thin films on reference substrates or on the 3D network?
- 6) Fig 2d: How was the thermal conductivity measured in detail? What technique has been used and what are the assumptions or the used model? What is the consideration about parasitic heatflow in the polymeric network? Ideally this can be investigated by Comsol simulations.
- 7) The power factor with 0,6-0,7 mW/cm²K² is comparable with sputtered thin films on planar substrate, but a factor of 5-8 times smaller than bulk materials. What is the filling fraction of the polymeric network and of the TE material per volume in reference to the device volume?
- 8) Regarding 2f, how many legs were measured in this device, 25?
- 9) The power density of the whole device (10 μW/g) should be translated to μW/cm² (device area) and compared with conventional devices and recent demonstrators?
- 10) The efficiency of the whole device (2f) was presumably calculated from the transport property of the thermoelectric material. These data should be compared with direct measurements of the thermoelectric conversion performance, by using a e.g. an ULVAC PEM system for the measurement of power output of the whole device.
- 11) For the final evaluation the internal device resistance of the thermoelectric device should be measured.

12) Regarding figure S6 the carrier concentration is around $1 \times 10^{20} / \text{cm}^3$ and decreases drastically with temperature. On the other hand, the carrier mobility is increasing with the temperature. I am very surprised about this behaviour. Regarding the n-type material a carrier concentration of $8,7 \times 10^{19} / \text{cm}^3$ is mentioned in the manuscript text at RT and the plot shows $8 \times 10^{21} / \text{cm}^3$. Also, for the n-type material the carrier concentration is reduced by one order of magnitude at higher temperature and the carrier mobility is also enhanced with temperature by one order of magnitude.

Reviewer #2 (Remarks to the Author):

In this work, V. Karthikeyan et al. report the fabrication of a 3D hollow architecture of thermoelectric materials, which is made of “bricks” including a carbon core and a $\text{Sb}_2\text{Te}_3/\text{Bi}_2\text{Te}_3$ shell. Such a structure is clarified to not only suppress heat stagnation in thermoelectric legs, but enhance mechanical robustness as well. To verify this viewpoint, the thermoelectric architectures are connected electrical in series and thermally in parallel to obtain a 10-couple thermoelectric device, which show a decent combination of mechanical/thermal stability and heat-to-electricity conversion efficiency. Given the conceptual novelty rationalized by this work, which might inspire future investigations towards energy-conversion techniques, I would like to recommend it for the publication in Nature Communications. However, there are some issues must be emphasized.

Major comments:

1. The concept of 3D architecture of thermoelectric materials proposed by this work is to some extent similar to previously reported by Nature Electronics|VOL 4|August 2021|579-587|. A reference of the previous work and a specific discussion of how this work advances beyond the previous work should be provided.
2. In terms of the synthesis of core-shell materials, a thermal evaporation method is implemented for coating on carbon framework. However, there can be significant inhomogeneity caused by such synthesis method, and more characterizations, e.g., the average thickness of coated layers, should be provided.
3. The authors only measure overall compression stress, which is mainly derived from the robustness of carbon framework. Supplementary characterizations (TEM) or measurements (electrical conductivity) of the structure after compressed or bended are required to solidly prove the materials are mechanically stable.
4. In measuring the conversion efficiency of assembled thermoelectric device, a fan is used to maintain cold-side temperature. However, this will induce turbulence or convection of air, which can have a giant influence for the hollow structure.
5. The authors claim the thermoelectric material is fabricated by additive manufacturing, which is not honest.

Minor comments:

6. "Here, we demonstrated for the first time, an approach to potentially enhance PCE meanwhile suppress the brittle failure in thermoelectric materials." Such a statement should be specified or just avoided, as there have been extensively reported thermoelectric materials with concurrently optimized figure-of-merit and mechanic properties.

7. "... we fabricate 3D architected TEGs that exhibited ultrahigh toughness per unit mass (30 MJ g⁻¹) ..." The unit of "MJ g⁻¹" is not matched with the definition of "toughness per unit mass".

8. "For any system, one-third of their total energy consumption is released as heat output ..." This statement is not rigorous. To my knowledge, most energy-conversion techniques release far more than 70% heat loss, yet, some Stirling engines can release less than 20% heat loss.

9. "Core-shell thermoelectric face-centered cubic microlattices are produced ..." According to Figure 1d, this is definitely not a face-centered structure.

10. In the section of "Thermoelectric properties of 3D core-shell Microlattices", the measurement method is ambiguous, and more details should be provided. For instance, for 2D (or even 3D) thermoelectric Sb₂Te₃ and Bi₂Te₃, the electrical and thermal conductivity are anisotropic, however, the measurement direction is unknown. Another issue is that it seems the electrical properties are measured on a free-standing Sb₂Te₃ or Bi₂Te₃ film, whereas the thermal properties are measured on a Sb₂Te₃ or Bi₂Te₃ film adhered on carbon framework.

11. "As per the Callaway's model, under relaxation time approximation, the phonon relaxation time purely depends on the grain size and thickness of the thin film." Here, the understanding of the Callaway's model is totally wrong, where under a relaxation time approximation, the phonon relaxation is not only dependent on intrinsic Umklapp and normal process, but also dependent on extrinsic scattering by participant defects, which is not merely a grain size in the case of a hollow 3D thermoelectric architecture.

12. In Figure 4a, what is the simulation accuracy, as there are some clear interfaces in the temperature profile of a bulk thermoelectric device. Moreover, in Figure 4b and 4c, which are presented to verify the simulation, the thermal boundary conditions are different to that used for simulation, based on which one cannot conclude that the simulation and experiment are mutually consistent. It is also observed that there are some hotter areas in Figure 4b and 4c, which probably refer to the frameworks. The authors should explain such a phenomenon.

13. "As shown in Figure 3b, the thermoelectric microlattices with carbonized polymer acting as its core, and thermoelectric thin film as its shell exhibits significantly enhanced compressive modulus of ~600 MPa and strength of ~30 MPa." These results are not agreed with Figure 3b.

14. "Throughout the experiment, the cold side of the TEG was found to remain at room temperature, demonstrating the ability of our microlattice TEGs to practically maintain significant thermal gradient and power output." Referring to comment 4, I am rather curious about what is the overall temperature profile of the 3D hollow architecture of thermoelectric materials.

Authors Response to Reviewer comments

Reviewer #1 (Remarks to the Author):

This publication on 3D architected Thermoelectric Devices shows an innovative concept for the fabrication of future thermoelectric devices. The manuscript will attract broad interest but the thermoelectric characterisation of the 3D device is not mature. The publication claims two high promises for the device with ultrahigh toughness and ultrahigh power efficiency. The first promise is proven by the mechanical testing of the device but the second claim of 10% conversion efficiency is not proven by this manuscript from my point of view. The following aspect are unclear and should be addressed in a revised manuscript:

Comment 1: Evaporation of metals and semiconductors is ideal of the thin film coatings of planar substrates. How homogenous and conformal is the coating in the center of the 3D architecture. The authors should prepare a cross-sectional cut and investigate this in detail by scanning electron microscopy.

Response: For the formation of uniform core-shell microlattice structure, we performed controlled optimization of the spin speed (rpm) of the microlattice sample holder and the deposition rate during the evaporation of Bismuth Telluride. Optimized rotation of 10 rpm and evaporation rate of 0.15 A/s was used to develop uniformity in the thermoelectric shell deposition over the partially carbonised polymer core under a high vacuum of 10^{-7} mbar. To prove the uniformity, we have made a cross-sectional cut of the core-shell microlattice structure as shown in Figure R1 below. Figure R1 depicts the uniformity in the deposition and the video file shows the uniformity in distribution of Bismuth Telluride shell over the polymer core. Our demonstration of uniformity in the shell coating by optimized deposition techniques over microlattice structures were reported in our previous works¹⁻³.

Figure R1 (a) Photograph of the uniformly coated TE shell over the polymer core (b) and (c) cross sectional SEM image of the fractured beam (d) exhibits the uniformity of the TE shell

over 3D microlattice core (e) video of the 3D core shell TE microlattices demonstrating the coating uniformity.

Comment 2: In figure 2 in many graphs it is not clear, which graph belongs to left and right scale?

Response: We thank the reviewer for spotting the misinterpretation, in the revised version we have modified the graph with arrows representing the corresponding axis in a more understandable format as shown below.

Figure R2 Thermoelectric properties of the architected TEGs. (a) and (b) Seebeck coefficient and electrical conductivity of the 3D p - Sb_2Te_3 and n - Bi_2Te_3 legs with respect to operating temperature. (c) Power factor of the 3D architected p - Sb_2Te_3 and n - Bi_2Te_3 thermoelectric legs with temperature. (d) Temperature-dependent thermal conductivity of p - and n -type thin films and the amorphous carbon core with respect to temperature. (e) Figure of merit (zT) p - Sb_2Te_3 , n - Bi_2Te_3 legs at different temperatures, evidencing the effect of partially carbonized core and architecture in enhancing the zT compared to previously reported works. (f) Maximum open-circuit voltage and specific power density of the 3D architected TEG. (h) Device efficiency and power density of the TEG with respect to the heat absorbed by the 3D core-shell microlattice structure.

Comment 3: Is the polymeric network structure completely electrical insulating? Does the polymeric network exhibit a Seebeck coefficient without the thermoelectric film? What is the resistance over temperature?

Response: The partially carbonized polymer core structure is a pure electrical insulator. Without the thermoelectric film (Bi_2Te_3), the microlattice structure doesn't exhibit thermoelectric effect hence Seebeck coefficient couldn't be obtained. The polymer core exhibits electrical resistance of $>15 \text{ M}\Omega$ over the temperature range of 300-550K. There is no electrical contribution from the polymer core as the material is partially carbonized and purely amorphous in nature, they only provide mechanical scaffold strength to the TE Shell on the surface.

Comment 4: Fig 2a and 2b: How was the electrical conductivity determined and what is the assumption about film thickness and film area on the network?

Response: Seebeck coefficient of the p- and n- type TE shell coated 3D microlattice were calculated from the slope of the voltage–temperature difference curve, the accuracy of the measurements was tested by comparing with the seebeck coefficients of their bulk ingots which is $\pm 5\%$.

The electrical conductivity of the TE shell coated 3D microlattice was calculated at each testing temperature by applying different current values to the sample and the resulting voltage values were recorded⁵. For the IV plot, electrical conductivity of one-micron thick TE film was calculated. For calculating the electrical conductivity, the area of TE shell was estimated by (1) initially measuring the geometrical dimensions of as fabricated TE microlattice via scanning electron microscope (SEM), such as strut/beam diameter, cubic unit cell length, TE film thickness, (2) creating a 3D reconstruction of the TE microlattice samples via a CAD software (SolidWorks) based on the experimentally measured dimensions to accommodate the deviations from initial CAD design before fabrication, and (3) obtaining the surface area and/or volume of the TE film calculated by the CAD software, which was also cross-checked by measuring the change in mass before and after coating. With the above mentioned measures, , the TE film thickness is found to be uniformly coated onto the 3D partially carbonized core, and the value used was the average film thickness measured from cross-section of struts at various regions within the sample.

Comment 5: Was the Hall effect determined on planar thin films on reference substrates or on the 3D network?

Response: The Hall effect for understanding the carrier dynamics of the deposited thermoelectric film was measured using the Van der Pauw four-probe method⁴, for this measurement we used the planar crossbar-shaped thin film of the same thickness as the thermoelectric shell ($1\mu\text{m}$) deposited on a silica substrate.

Comment 6: Fig 2d: How was the thermal conductivity measured in detail? What technique has been used and what are the assumptions or the used model? What is the consideration about parasitic heatflow in the polymeric network? Ideally this can be investigated by Comsol simulations.

Figure R3 Measured detector signal.

Response: As discussed in the manuscript, the thermal conductivity was measured using the Netzsch LFA 457 apparatus, here the laser/light flash method (Parker et al.,1961) is used to measure the thermal properties of the materials. In this technique, the lower surface of a test sample is first heated by a short energy pulse and the resulting temperature change on the upper surface of the sample is then measured with an infrared detector. The measured signal in

the detector is as represented in Figure R3, the higher the sample's thermal diffusivity, the higher the signal intensity. Using the half time ($t_{1/2}$), sample thickness (d) and the thermal diffusivity (α) its thermal conductivity (λ) can be calculated by means of the formula $\lambda(T) = \alpha(T) \cdot C_p(T) \cdot \rho(T)$, where C_p and ρ are specific heat and density of the material respectively. In our experiments, the thermal conductivity of the polymer core and thermoelectric materials were also measured using the LFA technique in Netzsch LFA 457 Hyperflash apparatus, as they support ultrafast recording of measurement data for thin film samples⁶. Carbonized circular polymer core samples coated with thermoelectric p- and n-type materials with a thickness of 10 μ m was prepared for the thermal conductivity measurements in the temperature range of 300-550 K.

Figure R4 COMSOL Multiphysics simulation for 3D Core-Shell TE microlattice structures and devices (a) Thermal conduction distribution of polymer core without TE shell and with p- and n-TE shell structure (b) and (c) simulated thermal gradient distribution of 3D thermoelectric whole device and one TE leg pair (d),(e) and (f) simulated voltage generation profile of 3D thermoelectric whole device and one TE leg pair (g) simulated power characteristics of 3D TE core-shell microlattice structure at a thermal gradient of 120°C .

The thermal conductivity results obtained were also validated with other measurement techniques such as thermoreflectance and 3-omega methods. The thermal conductivity of the 3D TE microlattice structure was estimated from the model illustrated by Shiva et al⁷. From the measurement results, it is well understood that the carbonized polymer core possesses

ultralow thermal conductivity than the thermoelectric shell p- and n-type materials, from which we considered the thermal contribution from the parasitic heat flow in the polymeric network to be negligibly small. However, to prove the power conversion performance through voltage profile and thermal profile distribution, we performed COMSOL simulation for the 3D thermoelectric microlattice structure.

Voltage profile and thermal gradient profile of single 3D TE pair and full 3D microlattice TE device is shown in Figure R4. As the reviewer suggested, we investigated the parasitic heat flow between the TE shell and polymeric core in the microlattice structure. In addition, simulations of heat flow with and without TE shell were performed as shown in Figure R4a. The results show that the heat flow in polymer core is almost negligible when compared to the heat flow with TE shell coated microlattice. Moreover, the voltage profile simulated with the 3D microlattice TE device validates our experimental results obtained.

Comment 7: The power factor with 0,6-0,7 mW/cm²K² is comparable with sputtered thin films on planar substrate, but a factor of 5-8 times smaller than bulk materials. What is the filling fraction of the polymeric network and of the TE material per volume in reference to the device volume?

Response: For the fabricated 3D core-shell microlattice TE legs, we report a maximum power factor of 7 and 6.3 $\mu\text{W cm K}^{-2}$ for p- and n-type materials respectively. However, in bulk samples, the power factor values are higher in the range of ~ 50 and $\sim 30 \mu\text{W/cmK}^2$ for p- Sb_2Te_3 and n- Bi_2Te_3 type materials respectively, owing to their larger carrier density and bulk electrical conductivity values⁸.

To estimate the filling fraction of the TE shell material and partially carbonized 3D core, we reconstructed the microlattice samples in a CAD software based on the measured dimensions of the sample, where the TE shell is essentially a hollow 3D shell covering the surface of the partially carbonized hybrid carbon core (Figure R5). With respect to a monolithic or bulk cubic sample or TE leg with the same external dimensions ($\sim 8 \text{ mm} \times 8 \text{ mm} \times 8 \text{ mm}$), the TE material has a volumetric filling fraction of $\sim 0.14\%$ (i.e., relative density, $\rho_{relative}$), while the partially carbonized hybrid carbon scaffold has a filling fraction of $\sim 7.4\%$. Note that $\rho_{relative}$ essentially represents the volumetric filling fraction of TE shell and hybrid carbon core when there is no spacing between the TE legs. The total volume of the device, V_{device} , is measured to be $\sim 16000 \text{ mm}^3$. Based on these values, the filling fraction, $V_{fraction}$, of the TE shell and hybrid carbon core per volume in reference to the device volume can be calculated as $V_{fraction} = N_{leg} \rho_{relative} V_{cubic} / V_{device}$, where N_{leg} represents the total number of TE legs (i.e., 20), $\rho_{relative}$ is the relative density of the TE shell and hybrid carbon core, V_{cubic} is the volume of the bulk cube with the same external dimensions as the lattice samples. Therefore, the $V_{fraction}$ of the TE shell with respect to device volume is $\sim 0.09\%$, while the $V_{fraction}$ of the hybrid carbon core with respect to device volume is $\sim 4.7\%$.

Figure R5 Relative densities of the TE shell, hybrid carbon core, and core-shell TE microlattice.

Comment 8: Regarding 2f, how many legs were measured in this device, 25?.

Response: We have fabricated the 3D core-shell microlattice TE device as shown in manuscript figure 2f using 10 p-n pairs of TE legs in series connection. The output power characteristics of this 10 p-n microlattice TE pairs are as shown in figure. Owing to the series connection, the internal resistance of the device is increased.

Question 9: The power density of the whole device ($10 \mu\text{W/g}$) should be translated to $\mu\text{W/cm}^2$ (device area) and compared with conventional devices and recent demonstrators?

Response: In the manuscript figure 2f, we have calculated the output power density (ω_{max}) of the device using the following relation:

$$\omega_{max} = \frac{(\Delta T)^2}{4L_{Leg}} PF_{avg}$$

where the ΔT is the temperature gradient between the hot and cold side, PF_{avg} is the average power factor, L_{Leg} is the length of the thermoelectric leg respectively. The calculated output power density ($\mu\text{W cm}^{-2}$) with respect to operating temperature is shown in the figure. The specific power density⁴ is defined as the power density per unit volume of the material expressed in $\mu\text{W g}^{-1}$ or $\mu\text{W cm}^{-3}$. We have reported a maximum output power density of $\sim 15 \mu\text{W cm}^{-2}$ and a maximum specific power density of $\sim 7 \mu\text{W g}^{-1}$ from our fabricated 3D microlattice TE device.

Table R1 Comparison of our 3D core-shell TE microlattice work with state of the art 3D printed TE devices and commercial TE devices.

Method	TE Materials	TE Leg Size	ΔT (K)	Voc (mV)	ω_{max} (W/cm ²)	Remarks
3D Direct Ink Writing Kim et al ⁹	Bi _{0.5} Sb _{1.5} Te ₃ (P) Bi ₂ Te _{2.7} Se _{0.3} (N)	350 μm	83	40	479 μ	Porous structure with poor mechanical strength and hard to sustain larger ΔT .

3D Extrusion printing Kim et al ⁴	Bi _{0.4} Sb _{1.6} Te ₃ (P) Bi ₂ Te _{2.7} Se _{0.3} (N)	10 mm	40	27	1.42 m	3D printed bulk TE leg structures and hard to sustain larger ΔT.
3D Shape Conformable Thermoelectric paint ¹⁰	Sb ₂ Te ₃ (P) Bi ₂ Te ₃ (N)	~25 mm	30	30	4m	Painting on 3D surface with thermoelectric ink for increasing thermal efficiency.
Commercial TEG	Sb₂Te₃(P) Bi₂Te₃(N)	~3 mm	230	0.5	2.1	Commercial Bulk TEG device with ~120 P-N leg pairs.
Dispenser Printing Chen et al ¹¹	Sb ₂ Te ₃ (P) Bi ₂ Te ₃ (N)	640 μm	20	320	75μ	2D planar device with 50 p-n leg pairs.
2D Thin film μTEG Vieira et al ¹²	Sb ₂ Te ₃ (P) Bi ₂ Te ₃ (N)	400 nm	35	210	3.3m	Co-evaporation fabricated 15 p-n TE leg pairs
Screen Printed flexible TEG Kim et al ¹³	Sb ₂ Te ₃ (P) Bi ₂ Te ₃ (N)	~500 μm	50	85	3.5m	Demonstrated wearable flexible TEG using glass fabric as scaffold
Wearable TEG Hong et al ¹⁴	Sb ₂ Te ₃ (P) Bi ₂ Te ₃ (N)	5 mm	10	1.75	25 μ	Wearable bulk TEG with thermoelectric pillars
Wearable TEG C.S Kim et al ¹⁵	Sb ₂ Te ₃ (P) Bi ₂ Te ₃ (N)	2.5 mm	10	85	38μ	Wearable bulk TEG with 160 pairs of commercial TEG legs
Screen printed flexible TEG Kim et al ¹⁶	Bi _{0.5} Sb _{1.5} Te ₃ (P) Bi ₂ Te _{2.7} Se _{0.3} (N)	~600 μm	26	700	6.32m	Ionized defect engineering processed screen-printed BiTeSe thick film device
3D Core-Shell TE microlattice (This work)	Sb₂Te₃(P) Bi₂Te₃(N)	8 mm	120	1100	1.45m	3D Core-shell TE devices were realized with larger efficiency due to controlled heat stagnation and larger thermal gradient in the TE legs.

Question 10: The efficiency of the whole device (2f) was presumably calculated from the transport property of the thermoelectric material. These data should be compared with direct measurements of the thermoelectric conversion performance, by using a e.g., an ULVAC PEM system for the measurement of the power output of the whole device.

Response: In the manuscript, the demonstrated power conversion efficiency of the 3D core-shell microlattice TE device was calculated using the standard procedure as elucidated below:

The power conversion efficiency of thermoelectric devices from heat energy to electricity can be defined as the ratio of energy provided to external load to heat absorbed at the hot junction. The overall maximum power efficiency (η_{device}) of thermoelectric devices is determined by:

$$\eta_{device} = \frac{\Delta T}{T_h} \frac{\sqrt{1+zT_M}-1}{\sqrt{1+zT_M}+T_c/T_h} \dots\dots\dots (1)$$

Where $\Delta T/T_h$ is the Carnot factor and the reduced efficiency depends on the zT , T_h and T_c .

Here, $zT_M = \frac{1}{T_H - T_C} \int zT dT$ is the average of the thermoelectric device figure of merit zT defined as

$$zT = \frac{(S_p - S_n)^2}{[\sqrt{\rho_n \kappa_n} - \sqrt{\rho_p \kappa_p}]^2} T \dots\dots\dots (2)$$

where S is the Seebeck coefficient, ρ is the electrical conductivity and κ is the thermal conductivity of p- and n-type materials depending on the absolute temperature T .

The efficiency of a thermoelectric device, as any other heat engine, is less than the Carnot engine efficiency:

$$\eta_{Carnot} = \frac{T_H - T_C}{T_C} \dots\dots\dots (3)$$

Indeed, the Carnot engine has the most efficient cycle for converting a given amount of thermal energy into work. In addition to the temperature difference, η_{device} is also related to the average of zT , which is related to the material properties of the p- and n-type materials used in the thermoelectric element. Moreover, as demonstrate in manuscript figure 2g the efficiency of TE device with different zT are compared with carnot engine efficiency.

On the other hand, the total power conversion efficiency with respect to heat absorbed in the 3D TE microlattice device is calculated using the following method¹⁷⁻¹⁹:

Maximum Power output, $P_{max} = \frac{V_{oc}^2}{4R_i}$

Power density, $\omega_{device} = \frac{P_{max}}{A_{module}}$

Heat conduction in the device, $Q_{out} = \kappa \cdot A \cdot \left(\frac{dT}{dx}\right)$

Total Power conversion efficiency of the device,

$$\eta = \frac{P_{max}}{Q_h} = \frac{P_{max}}{P_{max} + Q_{out}}$$

Where V_{oc} is the open circuit voltage, κ is thermal conductivity of the device, $\frac{dT}{dx}$ is the ratio of temperature difference and device thickness, Q_h is the absorbed heat and Q_{out} is the released heat respectively. Using this relation, we have performed this calculation using the heat flow and the maximum power output in the microlattice structure, as shown in the figure R6.

Figure R6 Power Conversion Efficiency and Power density with respect to heat absorbed by the 3D core-shell TE microlattice.

As suggested by the reviewer, the ULVAC PEM module evaluator also uses the same heat flow method for calculating the maximum power conversion efficiency. The results displayed are calculated using the same methods as the standard ULVAC PEM equipment's procedure.

Comment 11: For the final evaluation the internal device resistance of the thermoelectric device should be measured.

Response: Internal device resistance of the device with respect to temperature difference was measured and plotted in Figure R7, which shows the change in device internal resistance for different temperature gradient.

Figure R7 Internal Resistance of the 3D core-shell TE microlattice TEG.

Comment 12: Regarding figure S6 the carrier concentration is around $1 \times 10^{20}/\text{cm}^3$ and decreases drastically with temperature. On the other hand, the carrier mobility is increasing with the temperature. I am very surprised about this behaviour. Regarding the n-type material a carrier concentration of $8.7 \times 10^{19} / \text{cm}^3$ is mentioned in the manuscript text at RT and the plot shows $8 \times 10^{21} / \text{cm}^3$. Also, for the n-type material the carrier concentration is reduced by one order of magnitude at higher temperature and the carried mobility is also enhanced with temperature by one order of magnitude.

Response: Apologies for the misrepresentation of carrier concentration data, we have re-measured the carrier concentration and carrier mobility with respect to temperature and presented in Figure R8. At room temperature, the carrier concentration of p-Sb₂Te₃ and n-Bi₂Te₃ are $1.03 \times 10^{20}/\text{cm}^3$ and $9.8 \times 10^{19}/\text{cm}^3$ respectively. Here, the prepared thin film of Sb₂Te₃ and Bi₂Te₃ are polycrystalline nanograins structures which we illustrate in Figure S4 and S5 respectively. The increase in carrier mobility with temperature corresponds to the nanolattice scattering which is due to the potential barrier induced by the grain size effect following Petritz's mobility model^{20,21}. The updated Figure R8 demonstrates the temperature-dependent carrier concentration and mobility of p- and n-type TE thin films, where the carrier concentration decreases with temperature and mobility increases with temperature indicating the dominance of nanolattice scattering which in turn causes an increase in their electrical conductivity²²⁻²⁵.

Figure R8 Temperature-dependent Carrier Dynamics of p- and n-type TE shell.

Reviewer #2 (Remarks to the Author):

In this work, V. Karthikeyan et al. report the fabrication of a 3D hollow architecture of thermoelectric materials, which is made of “bricks” including a carbon core and a Sb₂Te₃/Bi₂Te₃ shell. Such a structure is clarified to not only suppress heat stagnation in thermoelectric legs, but enhance mechanical robustness as well. To verify this viewpoint, the thermoelectric architectures are connected electrical in series and thermally in parallel to obtain a 10-couple thermoelectric device, which show a decent combination of mechanical/thermal stability and heat-to-electricity conversion efficiency. Given the conceptional novelty rationalized by this work, which might inspire future investigations towards energy-conversion techniques, I would like to recommend it for the publication in Nature Communications. However, there are some issues must be emphasized.

Major comments:

Comment 1: The concept of 3D architecture of thermoelectric materials proposed by this work is to some extent similar to previously reported by Nature Electronics [VOL 4|August 2021|579-587]. A reference of the previous work and a specific discussion of how this work advances beyond the previous work should be provided.

Response: We agree with the reviewer that this work is to some extent related to a recently reported work published in Nature Electronics last year in which Kim et al⁹. used direct-ink writing to fabricate 3D thermoelectric cellular micro-architectures with a “woodpile-like” structure that can exhibit large temperature gradients and large power density. Nevertheless, the work in Nature Electronics is based on solid-beam structures composed entirely of (Bi,Sb)₂(Te,Se)₃-based thermoelectric materials, which are known to be highly brittle under mechanical compression (fracture strains below 5%). The intrinsic brittle nature of most high-

performance thermoelectric materials has been another longstanding issue which can hinder the widespread application of thermoelectric generators (TEGs).

In our work, we proposed a strategy to develop core-shell 3D TE microlattice architecture to suppress the intrinsic brittleness of thermoelectric materials under mechanical loading by incorporating a hybrid carbon core which not only possess superior strength but are also highly ductile (can withstand compressive strains greater than 50% with little to no fractures). Consequently, our core-shell thermoelectric microlattice can achieve both higher strengths and compressive strain compared to previously reported thermoelectric legs/materials. This enhanced strength and ductility or “tolerance” to deformation is crucial for the long-term reliability and mechanical robustness of these TEGs, as most thermoelectric legs are connected in series. Therefore, fracture in the thermoelectric material, which can be caused by external loads from various sources (e.g., vibrations), can result in short circuit that will render the entire device unusable. Furthermore, the use of a digital light projection (DLP)-based 3D printing system to create the 3D scaffold allows complex architectures to be fabricated with minimal to no additional support (e.g., overhanging structures/beams), which might be more challenging for direct ink writing (DIW)-based 3D printing to fabricate accurately without additional support material (e.g., drooping of overhanging beams prior to complete solidification) that will contribute to additional waste. This increased flexibility to fabricate complex geometries is beneficial in fabricating 3D architected thermoelectric devices with higher load-bearing efficiencies (specific strength) and/or deformability for a wider range of densities. Moreover, with our TE core-shell microlattice, we demonstrate sustaining a larger thermal gradient up to 120°C with high power density comparable with bulk devices. From our obtained results, we demonstrate more superior performance than the results reported in Kim et al., Nature Electronics 4, 579, 2021. Part of this discussion has been included in the revised version of the manuscript.

Comment 2: In terms of the synthesis of core-shell materials, a thermal evaporation method is implemented for coating on carbon framework. However, there can be significant inhomogeneity caused by such synthesis method, and more characterizations, e.g., the average thickness of coated layers, should be provided.

Response: For the formation of uniform core-shell microlattice structure, we performed controlled optimization of the spin speed (rpm) of the microlattice sample holder and the deposition rate during the evaporation of Bismuth Telluride. Optimized rotation of 10 rpm and evaporation rate of 0.15 A/s was used to develop uniformity in the thermoelectric shell deposition over the partially carbonised polymer core under a high vacuum of 10^{-7} mbar. To prove the uniformity, we have made a cross-sectional cut of the core-shell microlattice structure as shown in Figure R1 below. Figure R1 depicts the uniformity in the deposition and the video file shows the uniformity in distribution of Bismuth Telluride shell over the polymer core. The average thickness of the thermoelectric core is estimated from the cross section SEM analysis as $1.0 \pm 0.5 \mu\text{m}$.

Figure R5 (a) Photograph of the uniformly coated TE shell over the polymer core (b) and (c) cross sectional SEM image of the fractured beam (d) exhibits the uniformity of the TE shell over 3D microlattice core (e) video of the 3D core shell TE microlattices demonstrating the coating uniformity.

Comment 3: The authors only measure overall compression stress, which is mainly derived from the robustness of carbon framework. Supplementary characterizations (TEM) or measurements (electrical conductivity) of the structure after compressed or bended are required to solidly prove the materials are mechanically stable.

Response: We thank the reviewer for the suggestion. We have performed additional experiments where the 3D TE microlattices were subjected to different compressive strain levels (i.e., 10%, 25%, 50% and 75%) and simultaneously their change in electrical resistivity were measured after each strain levels. From our measurement, we observed that the change in electrical resistivity with respect to temperature is negligible in the range of ± 0.3 mOhm.cm for both p- and n-type respectively. Moreover, the change in resistivity is saturated over a compression stain of 50% as shown in the figure below. This mechanical characteristic of the TE shell arises from the TE thin film coated shell supported by the high deformation strength of the microlattice core. From these results we claim that the 3D core-shell microlattice TE device exhibits an ultrahigh toughness range.

Figure R9 Mechanical strain-dependent electric resistivity changes in 3D core-shell TE microlattice structure

Comment 4: In measuring the conversion efficiency of assembled thermoelectric device, a fan is used to maintain cold-side temperature. However, this will induce turbulence or convection of air, which can have a giant influence for the hollow structure.

Response: The output power characteristics of the fabricated 3D core-shell microlattice TE device was measured under different temperature gradients of 20, 40, 80 and 120°C respectively. To maintain the required temperature gradient, a dc heater was used on the hot side and the cold side was left open to ambient atmosphere, no forced convection techniques were employed on the cold side.

Moreover, the p-Sb₂Te₃ and n-Bi₂Te₃ thermoelectric legs are not hollow in structure. The legs are developed over a polymer core with a thermoelectric shell forming a 3D core-shell TE microlattice structure. Here, the 3D microlattice polymer core is electrically insulating (>15MΩ) which acts as a scaffold providing exceptional mechanical strength and ductility/deformability to the TE shell and also to sustain a larger temperature gradient naturally.

Comment 5: The authors claim the thermoelectric material is fabricated by additive manufacturing, which is not honest.

Response: We apologize for any misunderstanding with the use of the word “additive manufacturing” in the manuscript. One of the central ideas of our work is to leverage the geometrical flexibility of additive manufacturing technology to fabricate complex 3D architected microlattice structures which can serve as a potential replacement to traditional monolithic thermoelectric legs and increase the thermal gradient sustained in a TEG. Therefore, we are not claiming that the thermoelectric material is fabricated solely by additive manufacturing, but rather the core-shell microlattice which serves as thermoelectric legs in our architected TEGs possesses a complex 3D cellular geometry that was made using additive manufacturing. Nevertheless, we have removed the word “additive manufacturing” to avoid any potential confusion in the revised manuscript.

Comment 6: “Here, we demonstrated for the first time, an approach to potentially enhance PCE meanwhile suppress the brittle failure in thermoelectric materials.” Such a statement should be specified or just avoided, as there have been extensively reported thermoelectric materials with concurrently optimized figure-of-merit and mechanic properties.

Response: We have removed the statement “demonstrated for the first time” from our manuscript. Nevertheless, we would still like to further clarify our initial statement to avoid any misunderstanding. We agree with the reviewer that thermoelectric materials with concurrently optimized figure-of-merit and mechanical properties have been reported previously. However, such works mainly focused on thermoelectric materials with specific composition and/or manufacturing process. Furthermore, thermoelectric materials with simultaneously high specific compressive strength (exceeding structural alloys such as 316L stainless steel) and ductility at low densities have not been reported thus far to our knowledge owing to the well-known trade-off between strength and ductility in structural materials.

Thermoelectric materials with either enhanced strength or ductility (e.g., Ag₂S, InSe) are often inferior in terms of their thermoelectric performance (e.g., figure-of-merit, power conversion efficiency) compared with Bi₂Te₃ and SnSe that are brittle in nature. On the other hand, rather than focusing on a specific type of thermoelectric material, our work demonstrates a “strategy” to enhance both the mechanical (strength and ductility) and power conversion efficiency of any thermoelectric device by the incorporation of a 3D hybrid carbon micro-architected platform. We believe the approach presented in this work could potentially be implemented to TEGs with different material systems and partially decouple the relationship between mechanical and thermoelectric performance in TEGs.

Comment 7: “... we fabricate 3D architected TEGs that exhibited ultrahigh toughness per unit mass (30 MJ g⁻¹) ...” The unit of “MJ g⁻¹” is not matched with the definition of “toughness per unit mass”.

Response: We apologize for the use of the term “toughness per unit mass” in our manuscript to represent the “specific energy absorption” capabilities (Figure 3f) of our thermoelectric microlattices, which might confuse readers on its definition. We have thus replaced and just kept “specific energy absorption” to avoid any confusions or misunderstandings. Specific energy absorption refers to the energy absorbed per unit volume (MJ m⁻³ or J cm⁻³), which for microlattices, can be obtained by the total area under the stress-strain curve, divided by the density (kg m⁻³ or g cm⁻³) of the microlattices. Therefore, the unit for such a parameter can be expressed as J g⁻¹, which have been well-defined and used by other researchers to describe the balance between compressive strength, ductility/deformability, and density of 3D architected materials^{26–29}.

Comment 8: “For any system, one-third of their total energy consumption is released as heat output ...” This statement is not rigorous. To my knowledge, most energy-conversion techniques release far more than 70% heat loss, yet, some Stirling engines can release less than 20% heat loss.

Response: We thank and agree with the reviewer's comment that most of the internal combustion engine releases over 60% of the energy consumed as heat to the surroundings. We have modified the sentence as "For most of the system, over two-third of the energy consumption is released as heat output..."

Comment 9: "Core-shell thermoelectric face-centered cubic microlattices are produced ..." According to Figure 1d, this is definitely not a face-centered structure.

Response: With regards to referring the lattice unit cell used in our work as a "face-centered cubic" structure, we followed the nomenclature that have been previously established pertaining to strut-based lattice geometries.³⁰⁻³² On the other hand, we also acknowledge that some researchers might have used the same name of "face-centered cubic" structure to refer to other lattice geometries in their work, which can give rise to some confusion or inconsistencies in the naming system. To address this issue, we have revised our lattice as a "face-centered cubic (FCC)-like" and included an additional definition or description with regards to its construction (i.e., composed of diagonal struts from the edges of the cubic unit cell intersecting at the center of the cubic faces). In our work, we choose this type of architecture mainly owing to its "openness" which allows easy access to inner struts for homogenous coating while retaining sufficient stiffness and strength over conventional bending-dominated lattices such as the body-centered cubic lattices.

Comment 10: In the section of "Thermoelectric properties of 3D core-shell Microlattices", the measurement method is ambiguous, and more details should be provided. For instance, for 2D (or even 3D) thermoelectric Sb_2Te_3 and Bi_2Te_3 , the electrical and thermal conductivity are anisotropic, however, the measurement direction is unknown. Another issue is that it seems the electrical properties are measured on a free-standing Sb_2Te_3 or Bi_2Te_3 film, whereas the thermal properties are measured on a Sb_2Te_3 or Bi_2Te_3 film adhered on carbon framework.

Response: Anisotropy in the thermoelectric properties of 2D film was measured and didn't exhibit a wide variation for $1\mu\text{m}$. Moreover, only for the lakeshore Hall effect measurement to calculate carrier mobility and carrier concentration, a free-standing thin film of p- Sb_2Te_3 and n- Bi_2Te_3 were used as it is the requirement to have crossbar structure for the Van der Pauw method. Figure R10 below illustrates how the Seebeck and electrical conductivity measurements were performed.

Figure R10 Illustration of Seebeck Coefficient and Electrical Conductivity measurement.

Seebeck and the Electrical conductivity of the p-Sb₂Te₃ and n-Bi₂Te₃ were measured on a whole 3D core-shell microlattice TE leg structure by applying a temperature gradient between hot and cold sides as used in the standard equipment like Netzsch SBA 458 Nemesis. Electrical conductivities for the samples are measured using the standard 4-point setup at each temperature gradient. The resulting open circuit voltage were measured and the Seebeck coefficient was calculated at each temperature. The accuracy of the Seebeck coefficient and electrical conductivity measured are $\pm 5\%$ and $\pm 7\%$ respectively.

Comment 11: “As per the Callaway’s model, under relaxation time approximation, the phonon relaxation time purely depends on the grain size and thickness of the thin film.” Here, the understanding of the Callaway’s model is totally wrong, where under a relaxation time approximation, the phonon relaxation is not only dependent on intrinsic Umklapp and normal process, but also dependent on extrinsic scattering by participant defects, which is not merely a grain size in the case of a hollow 3D thermoelectric architecture.

Response: Here in our work, the fabricated thin films of p-Sb₂Te₃ and n-Bi₂Te₃ exhibits ultralow thermal conductivity values in the range between 0.27 and 0.24 W/mK at room temperature which is closer to the theoretical limit of thermal conductivity values as per Debye-Cahill model^{33,34}.

Under relaxation time approximation, the scattering mechanism of phonons depends on various scattering points which includes dislocations, point defects, phonon-phonon scattering, grain boundary scattering and thickness. Accordingly the total phonon relaxation time (τ_c) can be formulated as³⁴:

$$\tau_c^{-1} = \frac{c}{D} + \frac{c}{t} + A\omega^4 + B\omega^2 T \exp\left(-\frac{\theta_D}{3T}\right) + C\omega$$

Where D and t are the grain size and thickness of the thin films respectively. Depending on the grain size and film thickness, the phonon-phonon umklapp scattering is significantly affected, owing to this process, thermal conductivity of the samples approaches toward the minimum

thermal conductivity range. In this case, while we agree with the reviewer's comment that we should consider the influence of defect-phonon scattering, their dominance over the thermal conductivity compared to the grain size and film thickness is lower.

Comment 12: In Figure 4a, what is the simulation accuracy, as there are some clear interfaces in the temperature profile of a bulk thermoelectric device. Moreover, in Figure 4b and 4c, which are presented to verify the simulation, the thermal boundary conditions are different to that used for simulation, based on which one cannot conclude that the simulation and experiment are mutually consistent. It is also observed that there are some hotter areas in Figure 4b and 4c, which probably refer to the frameworks. The authors should explain such a phenomenon.

Response: We thank the reviewer for this comment. In Figure 4a, the clear interfaces in the temperature profile contour of the bulk thermoelectric leg are only caused by the visualization mode of the contour being discrete rather than related to the simulation accuracy. Changing the visualization mode from discrete to continuous shows a smooth transition in the temperature profile contour (Figure R11a), as verified in the temperature profile curve from the bottom surface to the top surface (Figure R11b). We have also changed the thermal boundary conditions of the simulation to match the experimental conditions to ensure both the simulation and experiment are mutually consistent (Figure R11a). In Figure 4b and c, we attribute the hotter areas in the infra-red images to the difference in thermal conductivity of the samples. For instance, in Figure 4b, the bulk material conducts heat slower than the microlattice sample, giving rise to hotter area near the bottom surface of the bulk sample. Similarly, the thermal conductivity of the N-type thermoelectric material is lower than that of the P-type material, eventually results in the area near the bottom surface of the N-type thermoelectric microlattice being hotter than the area at the bottom of the P-type sample (Figure 4c). This discussion has been added into the revised version of the manuscript.

Figure R11 (a) Simulated visualization of temperature profile from the bottom to the top surface comparison (b) Temperature profile variation with respect to structure height in bulk and 3D microlattice structures.

Comment 13: “As shown in Figure 3b, the thermoelectric microlattices with carbonized polymer acting as its core, and thermoelectric thin film as its shell exhibits significantly enhanced compressive modulus of ~600 MPa and strength of ~30 MPa.” These results are not agreed with Figure 3b.

Response: We thank the reviewer for pointing this out. The values written in our manuscript represent average values obtained across multiple samples (8 samples for each type of microlattice tested), while the stress-strain curves in Figure 3b were obtained from representative samples. We have remeasured the values in accordance with the stress-strain curve in Figure 3b and have corrected the values to compressive modulus of ~ 450 MPa and strength of ~ 35 MPa, respectively.

Comment 14: “Throughout the experiment, the cold side of the TEG was found to remain at room temperature, demonstrating the ability of our microlattice TEGs to practically maintain significant thermal gradient and power output.” Referring to comment 4, I am rather curious about what is the overall temperature profile of the 3D hollow architecture of thermoelectric materials.

Response: In order to prove the thermal profile of the 3D core-shell TE microlattice device, we performed a COMSOL simulation of heat conduction in the structure at maximum operating conditions, which exhibits the capability of maintaining ambient temperature as shown in Figure R4. Experimentally, our infrared images also demonstrate the ambient stability of the structure under any operating conditions. This phenomenon is observed owing to the prevention of heat stagnation in the microlattice structure compared to the conventional bulk structure.

Figure R6 COMSOL Multiphysics simulation for 3D core-shell TE microlattice structures and devices (a) Thermal conduction distribution of polymer core without TE shell and with p- and n-TE shell structure (b) and (c) simulated thermal gradient distribution of 3D thermoelectric whole device and one TE leg pair (d),(e) and (f) simulated voltage generation profile of 3D thermoelectric whole device and one TE leg pair. (g) Simulated power characteristics of 3D TE core-shell microlattice structure at a thermal gradient of 120°C.

References

1. Surjadi, J. U., Gao, L., Cao, K., Fan, R. & Lu, Y. Mechanical Enhancement of Core-Shell Microlattices through High-Entropy Alloy Coating. *Sci. Rep.* **8**, 5442 (2018).
2. Gao, L. *et al.* High-Entropy Alloy (HEA)-Coated Nanolattice Structures and Their Mechanical Properties. *Adv. Eng. Mater.* **20**, 1–8 (2018).
3. Surjadi, J. U., Feng, X., Zhou, W. & Lu, Y. Optimizing film thickness to delay strut fracture in high-entropy alloy composite microlattices. *Int. J. Extrem. Manuf.* **3**, 25101 (2021).
4. Kim, F. *et al.* 3D printing of shape-conformable thermoelectric materials using all-inorganic Bi₂Te₃-based inks. *Nat. Energy* **3**, 301–309 (2018).
5. SBA 458 Nemesis - Netzsch Analyses & Tests (<https://analyzing-testing.netzsch.com/en/products/seebeck-analyzer-sba/sba-458-nemesis>)
6. LFA 467 HyperFlash - NETZSCH Analyzing & Testing(<https://analyzing-testing.netzsch.com/en/products/thermal-diffusivity-and-conductivity/lfa-467-hyper-flash-light-flash-apparatus>).
7. Farzinazar, S., Schaedler, T., Valdevit, L. & Lee, J. Thermal transport in hollow metallic microlattices. *APL Mater.* **7**, 101108 (2019).
8. Nozariasbmarz, A. *et al.* Bismuth Telluride Thermoelectrics with 8% Module Efficiency for Waste Heat Recovery Application. *iScience* **23**, (2020).
9. Kim, F. *et al.* Direct ink writing of three-dimensional thermoelectric microarchitectures. *Nat. Electron.* **4**, 579–587 (2021).
10. Park, S. H. *et al.* High-performance shape-engineerable thermoelectric painting. *Nat. Commun.* **7**, 1–10 (2016).
11. Chen, A., Madan, D., Wright, P. K. & Evans, J. W. Dispenser-printed planar thick-film thermoelectric energy generators. *J. Micromechanics Microengineering* **21**, 104006 (2011).
12. Vieira, E. M. F. *et al.* High-Performance μ -Thermoelectric Device Based on Bi₂Te₃/Sb₂Te₃ p–n Junctions. *ACS Appl. Mater. Interfaces* **11**, 38946–38954 (2019).
13. Kim, S. J., We, J. H. & Cho, B. J. A wearable thermoelectric generator fabricated on a glass fabric. *Energy Environ. Sci.* **7**, 1959–1965 (2014).

14. Hong, S. *et al.* Wearable thermoelectrics for personalized thermoregulation. *Sci. Adv.* **5**, eaaw0536 (2022).
15. Kim, C. S. *et al.* Self-Powered Wearable Electrocardiography Using a Wearable Thermoelectric Power Generator. *ACS Energy Lett.* **3**, 501–507 (2018).
16. Kim, S. J. *et al.* Post ionized defect engineering of the screen-printed Bi₂Te_{2.7}Se_{0.3} thick film for high performance flexible thermoelectric generator. *Nano Energy* **31**, 258–263 (2017).
17. Nozariasbmarz, A. *et al.* High Power Density Body Heat Energy Harvesting. *ACS Appl. Mater. Interfaces* **11**, 40107–40113 (2019).
18. F-PEM | Atmospheric Thermoelectric Module Evaluation System (<https://showcase.ulvac.co.jp/en/products/heat-treatment-and-thermal-properties/thermoelectric-evaluation-device/atmosphere-thermoelectric-module/f-pem.html>).
19. Li, W. *et al.* Conformal High-Power-Density Half-Heusler Thermoelectric Modules: A Pathway toward Practical Power Generators. *ACS Appl. Mater. Interfaces* **13**, 53935–53944 (2021).
20. Micocci, G., Tepore, A., Rella, R. & Siciliano, P. Electrical properties of vacuum-deposited polycrystalline InSe thin films. *Sol. Energy Mater.* **22**, 215–222 (1991).
21. Zhang, D. H. & Ma, H. L. Scattering mechanisms of charge carriers in transparent conducting oxide films. *Appl. Phys. A* **62**, 487–492 (1996).
22. Rogacheva, E. I. *et al.* Growth and structure of thermally evaporated Bi₂Te₃ thin films. *Thin Solid Films* **612**, 128–134 (2016).
23. Delves, R. T., Bowley, A. E., Hazelden, D. W. & Goldsmid, H. J. Anisotropy of the Electrical Conductivity in Bismuth Telluride. *Proc. Phys. Soc.* **78**, 838–844 (1961).
24. George, J. & Pradeep, B. Preparation and properties of co-evaporated bismuth telluride [Bi₂Te₃] thin films. *Solid State Commun.* **56**, 117–120 (1985).
25. Hansen, N. Hall–Petch relation and boundary strengthening. *Scr. Mater.* **51**, 801–806 (2004).
26. Tancogne-Dejean, T., Spierings, A. B. & Mohr, D. Additively-manufactured metallic micro-lattice materials for high specific energy absorption under static and dynamic loading. *Acta Mater.* **116**, 14–28 (2016).
27. Yuan, S., Chua, C. K. & Zhou, K. 3D-Printed Mechanical Metamaterials with High Energy Absorption. *Adv. Mater. Technol.* **4**, 1800419 (2019).
28. Zhang, P. *et al.* Mechanical design and energy absorption performances of rational gradient lattice metamaterials. *Compos. Struct.* **277**, 114606 (2021).
29. Xue, R. *et al.* Mechanical design and energy absorption performances of novel dual

scale hybrid plate-lattice mechanical metamaterials. *Extrem. Mech. Lett.* **40**, 100918 (2020).

30. Panesar, A., Abdi, M., Hickman, D. & Ashcroft, I. Strategies for functionally graded lattice structures derived using topology optimisation for Additive Manufacturing. *Addit. Manuf.* **19**, 81–94 (2018).
31. Li, C. *et al.* Architecture design of periodic truss-lattice cells for additive manufacturing. *Addit. Manuf.* **34**, 101172 (2020).
32. Benedetti, M. *et al.* Architected cellular materials: A review on their mechanical properties towards fatigue-tolerant design and fabrication. *Mater. Sci. Eng. R Reports* **144**, 100606 (2021).
33. Agne, M. T., Hanus, R. & Snyder, G. J. Minimum thermal conductivity in the context of: Diffusion-mediated thermal transport. *Energy Environ. Sci.* **11**, 609–616 (2018).
34. Chiritescu, C., Mortensen, C., Cahill, D. G., Johnson, D. & Zschack, P. Lower limit to the lattice thermal conductivity of nanostructured Bi₂Te₃-based materials. *J. Appl. Phys.* **106**, (2009).

REVIEWER COMMENTS

Reviewer #2 (Remarks to the Author):

The authors have addressed all the raised questions. I suggest to accept this paper for publication.

Reviewer #3 (Remarks to the Author):

In this paper the authors present 3D architected Thermoelectric Devices which is an innovative concept for the fabrication of future thermoelectric devices. I agree to reviewer 1 that the manuscript will attract broad interest but the thermoelectric characterisation and discussion of the results of the 3D device should be improved.

Comments to the answers to reviewer 1:

1. C1: It is not convincing that there is uniform deposition on the μm scale in three dimensions if they rotate their structure only around one axis. In the video one can not tell how uniform the deposition really is, only that there seems to be deposition on all sides. It would be better to show a SEM cross-section images from the center of the cube not only from one surface.
2. C2: Here everything is fine for the figure mentioned by the reviewer (fig 2) but in Figure 4 the same problem still exists
3. C4: How was the IV recorded? Four points in a line? Or van der Pauw?
4. C5: Why was the film deposited on silica instead of a planar polymer substrate? The growth / micro structure of the film could be different.
5. C6: Why was a different TE film thickness used for thermal transport measurement? You argue that the parasitic heat flow through the polymer scaffold is negligible even though it has $\sim 33\%$ of the thermal conductivity of the TE films and a much larger cross section. With laser flash you get the diffusivity. How did you obtain the density and heat capacity to get the thermal conductivity? Did you consider the individual contributions (matrix, TE material, air)?
6. C9: It would be better to give the specific power density of the device in $\mu\text{W}/\text{cm}^2/\text{K}^2$ to make it easier to compare to other works.
7. C10: To obtain the efficiency the heat flux needs to be measured. The mini PEM uses a heat flux sensor. From the comment it seems that in your case the heat flux was calculated rather than measured? Details of the characterization of the efficiency should be added. It remains unclear what

was measure or calculated and how. Please provide the details of the measurement results and calculations?

Additional comments:

1. The leg resistance seems not to scale with the number of leg pairs. Additionally, the device resistance is quite high which might be related to contact resistances. A high contact resistance deteriorates the device efficiency. Did you evaluate the contact resistance? What is the influence of the contact resistance on the efficiency in your case?
2. In this context also details of the device fabrication should be presented. How did you contact the leg pairs?
3. Details of the simulations are missing. Please provide the used material properties and boundary conditions.

Authors Response to reviewer Comments

Comments 1: It is not convincing that there is uniform deposition on the μm scale in three dimensions if they rotate their structure only around one axis. In the video one cannot tell how uniform the deposition really is, only that there seems to be deposition on all sides. It would be better to show a SEM cross-section images from the center of the cube not only from one surface.

Response: For the formation of uniform core-shell microlattice structure, we performed controlled optimization of the spin speed (rpm) of the microlattice sample holder and the deposition rate during the evaporation of Bismuth Telluride. Moreover, to ensure uniform deposition over the shadowed area, repeated coatings in all 3D directions were performed and their overall TE coating uniformity were examined by cross sectional SEM images shown in figure R1.

Our demonstration of uniformity in the shell coating by optimized deposition techniques over microlattice structures were reported in our previous works¹⁻³. SEM images depicted in Figure R1 are taken from the sliced different beams from central area of the 3D microlattice structure. From this demonstration, it can be clearly understood that the formation of 3D Core-Shell Thermoelectric microlattice are uniform all over the carbonised structure.

Figure R1 SEM images of thermoelectric materials coated 3D microlattice structures (a) Lateral surface view (b) Sliced view of microlattice beam demonstrating uniform deposition (c-f) Cross-sectional view of uniformly deposited TE shell layer onto carbonised core structure.

Comments 2: Here everything is fine for the figure mentioned by the reviewer (fig 2) but in Figure 4 the same problem still exists.

Response: We have corrected the figure 4 as suggested.

Figure R2 Revised figure as commented by reviewer.

Comments 3: How was the IV recorded? Four points in a line? Or van der Pauw?

Response: IV characteristics for the p- and n-type samples were measured by 4-point method. IV characteristics under different temperature conditions are measured and their corresponding electrical conductivity were calculated⁴.

Comments 4: Why was the film deposited on silica instead of a planar polymer substrate? The growth / microstructure of the film could be different.

Response: Considering the sample requirements for the Hall measurements, the thermoelectric thin films were deposited over silica substrate for determining their carrier concentration. Moreover, the quality of the thin film growth was maintained with the help of a seed layer of the pristine material over the silica substrates.

Comments 5: Why was a different TE film thickness used for thermal transport measurement? You argue that the parasitic heat flow through the polymer scaffold is negligible even though it has ~33% of the thermal conductivity of the TE films and a much larger cross section. With laser flash you get the diffusivity. How did you obtain the density and heat capacity to get the thermal conductivity? Did you consider the individual contributions (matrix, TE material, air) ?

Response: For accuracy and to avoid thermal fluctuations in the thin film during the measurement, a TE film thickness of 10 μm was used for the thermal conductivity analysis.

For measuring the thermal conductivity of the TE thin film and the carbonised polymer structure, we used Netzsch LFA 467 Hyperflash apparatus. The Netzsch LFA 467 hyperflash apparatus⁵ measures the thermal diffusivity (α) in the samples and with a reference specimen specific heat C_p are determined with respect to the temperature. Using these measured values, thermal conductivity $\lambda(T) = \alpha(T) \cdot C_p(T) \cdot \rho$ were determined, and the bulk density ρ was measured using Archimedes principle.

Thermal conductivity of the partially carbonised polymer core and thermoelectric shell depicts that the polymer core contributes least to the thermal conduction. The 0.11W/m. K arises from the carbonised area of the polymer core; but the amorphous nature of the carbonised polymer microlattice cannot cause significant parasitic heat flow in the core-shell structure. Therefore, we argue that the parasitic heat flow in the polymer scaffold is negligible.

The data presented in the manuscript is the total thermal conductivity measured under nitrogen environment. To prove the distribution of thermal profile in the 3D TE structure under ambient environment, we also simulated the heat transfer in the structure as demonstrated by COMSOL simulation in supporting information figure S10.

Comments 6: It would be better to give the specific power density of the device in $\mu\text{W}/\text{cm}^2/\text{K}^2$ to make it easier to compare to other works.

Response: Specific power density is the power density per unit volume of the material expressed in $\mu\text{W g}^{-1}$ or $\mu\text{W cm}^{-3}$. We have reported a maximum output power density of $\sim 15 \mu\text{W cm}^{-2}$ and a maximum specific power density of $\sim 7 \mu\text{W g}^{-1}$ from our fabricated 3D microlattice TE device and not possible to represent in $\mu\text{W}/\text{cm}^2/\text{K}^2$. For easier comparison we made the comparison table of maximum output power density of various works reported as shown in supporting information table S1.

Comments 7: To obtain the efficiency the heat flux needs to be measured. The mini-PEM uses a heat flux sensor. From the comment it seems that in your case the heat flux was calculated rather than measured? Details of the characterization of the efficiency should be added. It remains unclear what was measure or calculated and how. Please provide the details of the measurement results and calculations?

Response: Heat flux applied to the hot side alumina plate of the 3D micro lattice TE device was calculated from the experimental measurement by using commercial heat flux sensor Hukseflux FHF05SC series thermal sensors. However, the active area of the device was calculated using the 3D micro lattice TE legs area. The demonstrated power conversion efficiency of the 3D core-shell microlattice TE device was calculated using the standard procedure as elucidated below:

The power conversion efficiency of thermoelectric devices from heat energy to electricity can be defined as the ratio of energy provided to external load to heat absorbed at the hot junction. The overall maximum power efficiency (η_{device}) of thermoelectric devices is determined by:

$$\eta_{device} = \frac{\Delta T}{T_h} \frac{\sqrt{1+zT_M}-1}{\sqrt{1+zT_M}+T_c/T_h} \dots\dots\dots (1)$$

Where $\Delta T/T_h$ is the Carnot factor and the reduced efficiency depends on the zT , T_h and T_c . Here, $zT_M = \frac{1}{T_H-T_C} \int zT dT$ is the average of the thermoelectric device figure of merit zT defined as

$$zT = \frac{(S_p-S_n)^2}{[\sqrt{\rho_n\kappa_n}-\sqrt{\rho_p\kappa_p}]^2} T \dots\dots\dots (2)$$

where S is the Seebeck coefficient, ρ is the electrical conductivity and κ is the thermal conductivity of p- and n-type materials depending on the absolute temperature T .

The efficiency of a thermoelectric device, as any other heat engine, is less than the Carnot engine efficiency:

$$\eta_{Carnot} = \frac{T_H-T_C}{T_c} \dots\dots\dots (3)$$

Indeed, the Carnot engine has the most efficient cycle for converting a given amount of thermal energy into work. In addition to the temperature difference, η_{device} is also related to the average of ZT , which is related to the material properties of the p- and n-type materials used in the thermoelectric element. Moreover, as demonstrate in manuscript figure 2g the efficiency of TE device with different zT are compared with carnot engine efficiency.

On the other hand, the total power conversion efficiency with respect to heat absorbed in the 3D TE microlattice device is calculated using the following method⁶⁻⁸:

Maximum Power output, $P_{max} = \frac{V_{oc}^2}{4R_i}$

Power density, $\omega_{device} = \frac{P_{max}}{A_{module}}$

Heat conduction in the device, $Q_{out} = \kappa \cdot A \cdot \left(\frac{dT}{dx}\right)$

Total Power conversion efficiency of the device,

$$\eta = \frac{P_{max}}{Q_h} = \frac{P_{max}}{P_{max} + Q_{out}}$$

Where V_{oc} is the open circuit voltage, κ is thermal conductivity of the device, $\frac{dT}{dx}$ is the ratio of temperature difference and device thickness, Q_h is the absorbed heat and Q_{out} is the released heat respectively. Using this relation, we have performed this calculation using the heat flow and the maximum power output in the microlattice structure, as shown in the manuscript figure 2h.

Comments 8: The leg resistance seems not to scale with the number of leg pairs. Additionally, the device resistance is quite high which might be related to contact resistances. A high contact resistance deteriorates the device efficiency. Did you evaluate the contact resistance? What is the influence of the contact resistance on the efficiency in your case?

Response: For the 3D TE microlattice device fabrication assembly, Ag-Sn solder pastes were used for the contact formation between 3D microlattice and Ni electrodes. Hence the contact resistance contribution is much lower than the TE materials resistance and their effect on the device efficiency cannot be observed. The maximum power output P_{max} is defined as:

$$P_{max} = \frac{V_{oc}^2}{4R_i}$$

Figure R3 (a) Total internal resistance of 3D TE device **(b)** Specific contact resistance of 3D p- and n-TE microlattice structures. Here the total device resistance R_i is defined as the sum of resistance of TE material (R_{TE}) solder paste (R_{Solder}), nickel metallic electrode (R_{metal}), electrical contact resistance between TE material and solder paste (R_{c1}) and electrical contact resistance between solder paste and Ni metal electrode (R_{c2}).

$$R_i = R_{TE} + 2R_{Solder} + 2R_{metal} + 2R_{c1} + 2R_{c2}$$

but the R_{Solder} , R_{metal} and R_{c2} can be neglected because they are considered to have very small resistances than the TE material. So, the main resistance affecting the maximum power output of our 3D TE microlattice device is: $R_i = R_{TE} + 2R_{c1}$. To estimate the contact resistance, we used the transfer length method⁹ (TLM) with respect to the change in temperature as shown in figure R3. From the results, it can be understood that the contribution of R_{TE} dominates the maximum power output and the power conversion efficiency of the 3D microlattice TE device.

Comments 9: In this context also details of the device fabrication should be presented. How did you contact the leg pairs?

Response: Now, we have included the device fabrication process in the experimental section of the revised manuscript. *3D Microlattice TE Device Fabrication:* From the thermally deposited p- and n-type 3D TE microlattices, a thermoelectric device with 10 pairs of p-n TE legs. The p-n legs are connected in series with nickel electrodes with the use of Ag-Sn solder paste with thin insulating ceramic plate as thermal contacts as shown in figure 1. Further, the fabrication 3D TE device assembly was annealed at 220°C to establish mechanical stability to the assembly by soldering the structure between the ceramic plates. Figure R4 below demonstrates the results of each fabrication steps starting from (a)“as-printed” 3D PEGDA microlattice, (b) partially carbonised 3D microlattice core, (c) p- and n-type thermoelectric shell deposited 3D microlattice, (d) p-n series assembled TE device and (e) fully fabricated 3D TE core-shell microlattice device.

Figure R4 (a) As- printed PEGDA 3D microlattice (b) Carbonised PEGDA structure (c) p- and n-type 3D Core-shell TE microlattice structure (d) Device assembly of 3D TE microlattice device (e) Complete fabricated 3D TE device.(Scale: 1 cm)

Comments 10: Details of the simulations are missing. Please provide the used material properties and boundary conditions.

Response: We performed a finite element study using COMSOL Multiphysics for 3D architected TE generators (Figure S10). Our model calculates the thermal gradient profile of the 3D core-shell TE microlattice structures and full TE device integrated in between a pair of flat alumina plates. During the simulation the cold side and the ambient air was maintained at 25°C, with natural convective heat transfer coefficient of air as 10 W m² K⁻¹. On the basis of microlattice geometry and material properties the internal electrical resistance was estimated as 0.2 kOhm per TE pair at room temperature. The device geometry was maintained as the same experimented TE module dimensions with 10 TE leg pairs with height of 8 mm. Simulation for analysing the difference in thermal conduction with and without TE shell coating over the polymer core was performed as shown in Figure S9a, which demonstrates a negligible parasitic heat transfer between polymer core and TE shell of the 3D microlattice. Figure S9b and S9c demonstrates the thermal gradient distribution over the device at ΔT of 120°C. Simulated thermoelectric voltage distribution profile of the 3D device is shown in Figure S9d, S9e and S9f. The power generation at maximum temperature gradient of 120°C are shown in Figure S9g. We have already discussed the details of simulations in the manuscript supporting information section S5.

Reference

1. Surjadi, J. U., Gao, L., Cao, K., Fan, R. & Lu, Y. Mechanical Enhancement of Core-Shell Microlattices through High-Entropy Alloy Coating. *Sci. Rep.* **8**, 5442 (2018).
2. Gao, L. *et al.* High-Entropy Alloy (HEA)-Coated Nanolattice Structures and Their Mechanical Properties. *Adv. Eng. Mater.* **20**, 1–8 (2018).
3. Surjadi, J. U., Feng, X., Zhou, W. & Lu, Y. Optimizing film thickness to delay strut fracture in high-entropy alloy composite microlattices. *Int. J. Extrem. Manuf.* **3**, 25101 (2021).
4. SBA 458 Nemesis - Netzsch Analyses & Tests (<https://analyzing-testing.netzsch.com/en/products/seebeck-analyzer-sba/sba-458-nemesis>)
5. LFA 467 HyperFlash - NETZSCH Analyzing & Testing(<https://analyzing-testing.netzsch.com/en/products/seebeck-analyzer-sba/sba-458-nemesis>)

[testing.netzsch.com/en/products/thermal-diffusivity-and-conductivity/lfa-467-hyper-flash-light-flash-apparatus](https://www.netzsch.com/en/products/thermal-diffusivity-and-conductivity/lfa-467-hyper-flash-light-flash-apparatus)).

6. Nozariasbmarz, A. *et al.* High Power Density Body Heat Energy Harvesting. *ACS Appl. Mater. Interfaces* **11**, 40107–40113 (2019).
7. F-PEM | Atmospheric Thermoelectric Module Evaluation System (<https://showcase.ulvac.co.jp/en/products/heat-treatment-and-thermal-properties/thermoelectric-evaluation-device/atmosphere-thermoelectric-module/f-pem.html>).
8. Li, W. *et al.* Conformal High-Power-Density Half-Heusler Thermoelectric Modules: A Pathway toward Practical Power Generators. *ACS Appl. Mater. Interfaces* **13**, 53935–53944 (2021).
9. Kim, Y. *et al.* Practical evaluation of electrical contact resistance of thermoelectric legs at high operation temperature. *J. Mater. Sci.* **30**, 14112-14119 (2019).

REVIEWERS' COMMENTS

Reviewer #3 (Remarks to the Author):

The authors have answered the open questions. I recommend that the manuscript be accepted for publication.